



# Sources and cycling of nitrogen in a New England river discerned from nitrate isotope ratios

Veronica R. Rollinson[1], Julie Granger[1], Sydney C. Clark[2], Mackenzie L. Blanusa[1], Claudia P. Koerting[1], Jamie M.P. Vaudrey[1], Lija A. Treibergs[1,4], Holly C. Westbrook[1,3], Catherine M. Matassa[1], Meredith K. Hastings[2], Craig R. Tobias[1]

[1]Department of Marine Sciences, University of Connecticut, Groton, 06340, USA
[2]Department of Earth, Environmental and Planetary Sciences, Brown University, Providence, 02912, USA
[3]School or the Earth, Ocean and Environment, University of South Carolina, Columbia, 29208, USA
[4]Adirondack Watershed Institute, Paul Smith's College, Paul Smith's, 12970, USA

*Correspondence to*: Veronica R. Rollinson (veronica.rollinson@uconn.edu)

## Abstract

Coastal waters globally are increasingly impacted due to the anthropogenic loading of nitrogen (N) from the watershed. In order to assess dominant sources of N contributing to the eutrophication of the Little Narragansett Bay estuary in New England, we carried out an annual study of N loading from the Pawcatuck River. We conducted weekly monitoring of nutrients and nitrate ($NO_3^-$) isotope ratios ($^{15}N/^{14}N$, $^{18}O/^{16}O$ and $^{17}O/^{16}O$) at the mouth of the river and from the larger of two Waste Water Treatment Facilities (WWTFs) along the estuary, as well as seasonal along-river surveys. Our observations reveal a direct relationship between N loading and the magnitude of river discharge, and a consequent seasonality to N loading into the estuary – rendering loading from the WWTFs and from an industrial site upriver more important at lower river flows during warmer months, comprising ~23 % and ~18 % of N loading, respectively. Riverine nutrients derived predominantly from deeper groundwater and the industrial point source upriver during low base flow in summer, and from shallower groundwater and surface flow at higher river flows during colder months. Loading of dissolved organic nitrogen appeared to increase with river discharge, ostensibly delivered by surface water. The $NO_3^-$ associated with deeper groundwater had higher $^{15}N/^{14}N$ ratios than shallower groundwater, consistent with the expectation fractionation due to partial denitrification. Along-river, $NO_3^-$ $^{15}N/^{14}N$ ratios showed a correspondence to regional land use, increasing from agricultural and forested catchments to the more urbanized watershed downriver, with the agricultural and urbanized portions of the watershed contributing disproportionately to total N loading. Corresponding $NO_3^-$ $^{18}O/^{16}O$ ratios were lower during the warm season, a dynamic that we ascribe to increased biological cycling in-river. The $^{18}O/^{16}O$ isotope ratios along-river were consistent with the notion of nutrient spiraling, reflecting $NO_3^-$ input from the watershed and in-river nitrification and its coincident removal by biological consumption. Uncycled atmospheric $NO_3^-$, detected from its unique mass-independent $NO_3^-$ $^{17}O/^{16}O$ vs. $^{18}O/^{16}O$ fractionation, accounted for < 3 % of riverine $NO_3^-$, even at elevated discharge. We explore the implications of our findings for the mitigation of eutrophication in Little Narragansett Bay.



## 1. Introduction

Human activities have resulted in a substantial increase in the delivery of nutrients from terrestrial to aquatic and marine systems (Gruber and Galloway, 2008). In marine systems, increased loading of reactive nitrogen (N) has resulted in coastal eutrophication, engendering the loss of valuable nearshore habitat such as seagrass beds and oyster reefs, depletion of dissolved oxygen (creating so-called "dead zones"), and increased frequency and severity of algal blooms – including toxic brown and red tides causing fish kills (Heisler et al., 2008). In densely populated areas like the northeast United States, excess anthropogenic nitrogen loads originate from Waste Water Treatment Facilities (WWTFs), septic systems, industrial discharge, fertilizer applied to turf and agricultural lands, and atmospheric sources from industry and fossil fuel use (Valiela et al., 1997; McClelland et al., 2003, Latimer and Charpentier, 2010). The pervasive degradation of coastal marine ecosystems is alarming and of significant concern to coastal communities worldwide.

The transfer of nutrients from land to the coast is facilitated by rivers, which constitute an effective pipeline that collects nutrients from the watershed, ultimately discharging these to the coast. The mitigation of estuarine eutrophication thus relies on identifying primary sources of nutrients to riverine systems. Nutrients are fundamentally delivered to rivers from non-point sources: from waters entering the river via surface runoff, sub-surface groundwater in the unsaturated zone, and groundwater within the water table. Nutrients also enter rivers from point sources, including WWTFs as well as industrial discharge, which can dominate N loading in urbanized watersheds (Howarth et al., 1996). The nutrient loads contained in surface and deeper groundwater entering rivers differ markedly depending on land use. In temperate pristine systems, soil and groundwater concentrations are generally low, with reactive N originating from atmospheric deposition, biological $N_2$ fixation in soils, and from N in rocks and minerals (Hendry et al. 1984; Holloway et al. 1998; Morford et al., 2016). Higher concentrations of reactive N are found in waters draining agricultural and urbanized areas (Dubrovsky et al., 2010; Baron et al., 2013).

The N loaded to the watershed is partially attenuated through biological cycling in soils and aquifers. Specifically, organic N is degraded to reduced N species that are oxidized (nitrified) to nitrate ($NO_3^-$) in oxygenated zones of groundwater. $NO_3^-$ is otherwise removed from anoxic groundwater by denitrification, reduced to inert $N_2$. Reactive N is further cycled and attenuated in-river: The hyporheic zone, where groundwater interchanges with stream and river water, creates a complex



environment that can stimulate nitrification and denitrification, as oxic and anoxic pockets exist in close proximity (Sebilo et al., 2003; Harvey et al., 2013). Reactive nitrogen can be further attenuated by benthic denitrification within the river channel (Sebilo et al., 2003; Kennedy et al., 2008;

Mulholland et al., 2008).

Identifying sources of N to rivers can be difficult due to the expanse and heterogeneity of the watershed, the long integration time of deeper groundwater, and the degree of biological N cycling in groundwater and in-river. While measurements of N concentrations along the river channel in relation to regional land use can offer insights in this regard, N sources can be further resolved using

complementary measurements of the naturally occurring N and oxygen (O) isotope ratios of riverine $NO_3^-$ ($^{15}N/^{14}N$ and $^{18}O/^{16}O$, respectively). Henceforth, we express the isotope ratios in delta notation:

$$\delta\ (‰) = \left(\frac{isotope\ ratio\ of\ sample}{isotope\ ratio\ of\ reference} - 1\right) \times 1000 \qquad (1)$$

The reference for $\delta^{15}N$ is $N_2$ in air, and for $\delta^{18}O$ is Vienna Standard Mean Ocean Water (VSMOW). The N and O isotope ratios of $NO_3^-$ provide constraints on N sources and cycling in part because respective

N sources cover discrete ranges of $\delta^{15}N$ and $\delta^{18}O$ values (Kendall et al. 2007). Reactive N species from atmospheric deposition, biological $N_2$ fixation, and industrial $N_2$ fixation share overlapping ranges of $\delta^{15}N$ values (≤ 0 ‰), which differ appreciably from those of livestock and human waste (8 - 25 ‰; Kendall, 1998; Böhlke, 2003; Xue et al., 2009). In contrast, the $\delta^{18}O$ signatures of atmospheric $NO_3^-$ (60 – 80 ‰) are distinct from those of industrial $NO_3^-$ (~25 ‰) and $NO_3^-$ produced by nitrification (≤

1 ‰; Boshers et al. 2019 *and references therein*). Atmospheric $NO_3^-$ is further distinguished by a mass-independent $\delta^{17}O$ vs. $\delta^{18}O$ fractionation that is not manifest in industrial and biological $NO_3^-$ (Savarino and Thiemens, 1999).

The isotope ratios of $NO_3^-$ also provide constraints on N cycling because N and O isotopologues are differentially sensitive to respective biological N transformations (*reviewed by* Casciotti, 2016),

implicating different mass balance considerations within the N cycle that permit differentiation of N sources from cycling. Briefly, in riverine systems where $NO_3^-$ is the dominant N pool, $\delta^{15}N_{NO3}$ integrates across values of reactive N delivered from the watershed, minus $NO_3^-$ removed by benthic denitrification (if associated with N isotopic fractionation; Sebilo et al., 2003). Values of $\delta^{15}N_{NO3}$ are additionally sensitive to isotopic fractionation due to internal cycling in-river – assimilation and

remineralization to $NO_3^-$ via nitrification – in systems where riverine N is otherwise partitioned comparably between oxidized and reduced pools (*i.e.*, $NO_3^-$ vs. ammonium and particulate N; Sebilo





et al., 2006). Riverine $\delta^{18}O_{NO3}$, in turn, integrates across values of exogenous $NO_3^-$ delivered to the river from the watershed and from atmospheric deposition, those of $NO_3^-$ produced in-river by nitrification, minus the $NO_3^-$ lost concurrently to denitrification and assimilation (*see* Sigman et al.

2019). Interpreted in tandem, $NO_3^-$ N and O isotopologue ratios thus offer complementary constraints to identify important source terms and characterize inherent cycling.

Here we present a study of annual N loading from the Pawcatuck River to the Little Narragansett Bay in southern New England (U.S.A.), wherein we exploit measurements of the N and O isotope ratios of riverine $NO_3^-$ to draw inferences on dominant N sources from the watershed and on riverine

N cycling. The site is heavily impacted by nitrogen loading as evidenced by the history of the habitat: Vast seagrass beds of *Zostera marina* (eelgrass) historically established in Little Narragansett Bay were overtaken in the early 1990's by extensive mats of filamentous macroalgae dominated by the *Cladophoraceae* clade, whose substantial biomass has been linked to frequent events of night-time hypoxia in the bay's shallow-water coves (Dodds and Gudder. 1992; Dillingham et al., 1993; D'avanzo

and Kremer 1994; Tiner et al., 2003; Berezina and Golubkov, 2008; National Water Quality Monitoring Council, 2020). The evident eutrophication of the estuary has raised questions regarding the magnitude of N loading from the Pawcatuck River and from the local WWTFs, whose respective contributions must be assessed in order to devise targets for mitigation. To this end, we conducted weekly monitoring of nutrients and $NO_3^-$ isotopologue ratios at the mouth of the Pawcatuck River

and of nutrients discharged from the larger of two WWTFs along the estuary, as well as parallel measurements of samples collected from seasonal along-river surveys. Utilizing $NO_3^-$ isotopologue ratios to identify N sources has immediate local implications for management of the watershed, allows for extrapolation to similar watersheds throughout the temperate zone, and most importantly, isolates the seasonal and flow-dependent nature of N cycling within a riverine system transitioning

to an estuarine system. This last finding has direct relevance to water quality modeling efforts in temperate estuaries.

## 2. Methods

### 2.1 Site Description



The Pawcatuck River watershed (~760 km$^2$) is located predominantly in the state of Rhode Island

(RI) with as small portion in eastern Connecticut (Figure 1). The river originates at Worden Pond in

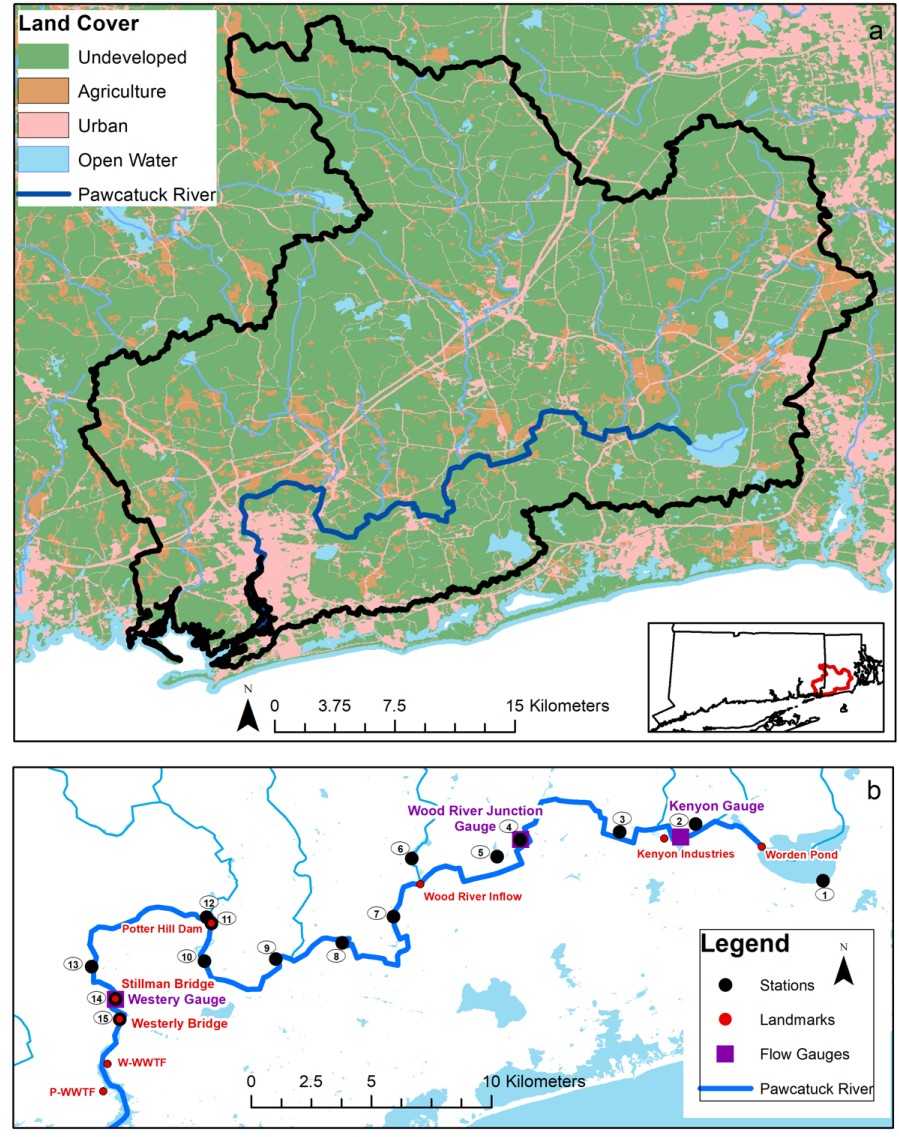

Figure 1. Map of (a) the Pawcatuck River watershed (URIEDC_RIGIS, 2019) and associated land use (U.S. Geological Survey, 2011), and (b) sampling locations and discharge gauges along the Pawcatuck River (U.S. Geological Survey, 2005; U.S. Census Bureau, 2017).

Wakefield, RI, and extends 47 km southwest to Westerly, RI. It is joined by the Wood River, which

originates in northern RI and runs 29 km south to Wood River Junction. The drainage basin is mostly





flat, hosting terrain with forests and wetlands (73 %) and relatively low human population (~56,400; based on a dasymetric analysis of the 2010 U.S. Census Bureau population data in the watershed;

Vaudrey et al., 2017) – owing in part to state land trust holdings that protect ~22 % of the watershed in RI from development (Dillingham et al. 1992; U.S. Geological Survey, 2011). Agricultural areas comprise 8 % of land use (U.S. Geological Survey, 2011) and are mostly located in the upper watershed, which hosts a number of turf farms: In 2005, Washington county – where the Pawcatuck River originates – was noted as having the highest density of turf farms in the United States (U.S.

Environmental Protection Agency, 2005). Urbanized and developed land usage comprises 13 % of the total watershed with the majority of the urban areas concentrated on the lower 19 km portion of the river, between Bradford and Westerly (U.S. Geological Survey, 2011).

Three nutrient discharge permits are allotted along the river by the RI Department of Environmental Management (RI DEM; Figure 1): Kenyon Industries, a fabric processing plant, is

located approximately 7 km downstream from Worden Pond. Two WWTFs discharge into the estuary and are located 1 km downstream of the Westerly Bridge, approximately 47 km downstream of Worden Pond.

2.2 *Sample collection*

We conducted four distinct sampling regimens: (a) weekly river samplings at the mouth of the

river, (b) weekly WWTF effluent samplings, (c) seasonal along-river surveys, and (d) rainwater samplings. We collected weekly river samples (a) from January 10, 2018 through to January 12, 2019 at two sites: The Stillman Bridge near the mouth of the freshwater portion of the river, and ~1 km downstream at the Westerly Bridge, which marks the limit of seawater intrusion (Figure 1, S1). (b) We obtained samples of wastewater treatment effluent collected weekly at the Westerly Waste

Water Treatment Facility (W-WWTF) from June 6, 2018 to May 22, 2019. (c) We conducted three seasonal along-river surveys on May 21, 2018, November 9, 2018, and March 12, 2019 at 15 discrete sampling stations between Worden Pond and the Westerly Bridge. Additionally, we performed a highly resolved sampling (approximately every 0.75 km) of the lower river from Potter Hill Dam (Station 11) to Westerly (Station 15) aboard kayaks in May 2017 (Figure 1). (d) We collected rainwater

samples following rain events from a rooftop collector at the Avery Point campus in Groton, CT (approximately 18 km west of the Pawcatuck River), from September 6, 2018 to December 2, 2018, in order to define regional $NO_3^-$ isotopic endmembers.



Weekly samplings at the Stillman and Westerly bridges occurred around sunrise, before the onset of photosynthetic activity, whereas along-river samples were collected sequentially from sunrise to mid-day. During each sample collection, river temperature and dissolved oxygen concentrations were measured *in situ* with a Thermo Orion Star A123 portable dissolved oxygen meter. At each site, river water was collected at ~0.5 m depth with a Van Dorn bottle and transferred into a 5 L carboy for transport, on ice, back to the laboratory for processing. In the laboratory, the conductivity of each sample was measured with an Oakton CON 450 conductivity meter. Sub-samples for analyses of dissolved nutrient and $NO_3^-$ isotope ratios were filtered through pre-combusted 25 mm GF/F glass fiber filters and collected in acid washed polypropylene bottles, then stored at -20˚C pending analysis. The filters were placed in pre-combusted aluminum foil and frozen at -20˚C in preparation for particulate nitrogen isotope ratio analyses. Samples for chlorophyll-a analysis were similarly collected onto 25 mm GF/F filters.

The weekly effluent samples at the Westerly WWTF were collected by facility personnel into 0.5 L acid-washed polypropylene bottles and frozen pending monthly pick-ups by our team. Two types of samples were collected on a weekly basis: grab and composite samples. Grab samples correspond to treated effluent collected prior to its release to the river, while composite samples are effluent collected continually over a 24-hour period, thus providing a concentration-weighted daily average. In the laboratory, samples for nutrient analysis were thawed and filtered through a 25 mm GF/F filter and frozen at -20 ˚C pending analysis. Samples for particulate N analysis were not collected from the WWTF.

Rainwater samples were collected into trace-metal-clean 1-L Teflon bottles outfitted with a glass funnel to create a vapor lock preventing evaporation. These samples were stored unfiltered at -20˚C pending nutrient and $NO_3^-$ isotope ratio analyses.

### 2.3 *Nutrient analyses*

The $NO_3^-$ concentration, $[NO_3^-]$, in river and WWTF samples was measured by conversion to nitric oxide in a hot Vanadium III solution followed by detection on a chemiluminescent NOx analyzer (Teledyne[TM]; Braman and Hendrix, 1989). Incident nitrite in the samples was first reacted with Griess reagents (Strickland and Parsons, 1972) before injection into the hot Vanadium (III) solution in order to detect $NO_3^-$ only. The concentration of nitrite, $[NO_2^-]$, in river samples was measured by conversion to nitric oxide in hot iodine solution, followed by detection on the chemiluminescent $NO_x$ analyzer





(Garside, 1982). For the rainwater samples, [$NO_3^-$] and [$NO_2^-$] were measured on a SmartChem discrete nutrient autoanalyzer (Unity Scientific™) using standard protocols adapted for the

SmartChem (Strickland and Parsons, 1972; U. S. Environmental Protection Agency, 1993b; 4500-$NO_2^-$, 2018; 4500-$NO_3^-$, 2018). Concentrations of ammonium, [$NH_4^+$], and phosphate, [$PO_4^{3-}$], in river and WWTF samples were measured on a SmartChem autoanalyzer using standard protocols (Murphy and Riley, 1962; Strickland and Parsons, 1972; U.S. Environmental Protection Agency, 1978, 1993; 4500-NH3, 2018; 4500-P, 2018).

The concentration of total dissolved nitrogen, [TDN], in filtered river and WWTF samples was measured by persulfate oxidation to $NO_3^-$, then measured via chemiluminescent NOx analyzer as described above (Sólorzano and Sharp, 1980; Knapp et al, 2005). The persulfate reagent was first recrystallized following protocol by Grasshoff et al. (1999). A ratio of sample to reagent of 5 to 10 was used in the oxidations. Reagent blanks accounted for ≤ 0.3 % of the TDN signal. The incident

concentration of dissolved organic nitrogen, [DON] was calculated as the difference between [TDN] and dissolved inorganic nitrogen, [DIN], where [DIN] = [$NO_3^-$] + [$NO_2^-$] + [$NH_4^+$].

2.4 *Chlorophyll-a analyses*

Chlorophyll-a was extracted from duplicate 25 mm GF/F filter samples in 5 mL of 90 % acetone, incubated overnight at -20˚C and quantified by florescence detection on a Turner Trilogy Laboratory

Fluorometer (Arar and Collins, 1997).

2.5 *$NO_3^-$ Isotope ratio analyses*

The nitrogen and oxygen isotope ratios of $NO_3^-$, $^{15}N/^{14}N$, $^{18}O/^{16}O$, and $^{17}O/^{16}O$ were analyzed using the denitrifier method in samples where [$NO_3^-$] ≥ 1.5µM (Sigman et al. 2001; Casciotti et al, 2002; Kaiser et al. 2007). Briefly, $NO_3^-$ was converted quantitatively to a nitrous oxide ($N_2O$) analyte by

denitrifying bacteria that lack a terminal reductase (*Pseudomonas chlororaphis* f. sp. *aureofaciens*; ATCC® 13985™), followed by analysis of the $N_2O$ product at the University of Connecticut on a Thermo Delta V GC-IRMS prefaced with a custom-modified Gas Bench II device with two cold traps and a PAL autosampler (Casciotti et al., 2002). The $NO_3^-$ $^{17}O/^{16}O$ in rainwater (as well as $^{18}O/^{16}O$) was similarly analyzed by bacterial conversion to $N_2O$, followed by pyrolysis in a gold tube to $N_2$ and $O_2$ and analysis

on a Thermo Delta V GC-IRMS at Brown University (Kaiser et al. 2007).

Coupled $\delta^{15}N_{NO3}$ and $\delta^{18}O_{NO3}$ analyses at UConn and Brown University were calibrated from parallel analyses of $NO_3^-$ reference materials USGS-34 ($\delta^{15}N$: -1.8 ‰ vs. air; $\delta^{18}O$: -27.9 ‰ vs. VSMOW)





and IAEA-N3 ($\delta^{15}$N: +4.7 ‰ vs. air; $\delta^{18}$O: +25.6 ‰ vs. VSMOW). Samples were analyzed in triplicate among two or more batch analyses. Reproducibility averaged 0.2‰ for $\delta^{15}$N$_{NO3}$ and 0.3‰ for $\delta^{18}$O$_{NO3}$.

Coupled analyses of $\delta^{18}$O$_{NO3}$ and $\delta^{17}$O$_{NO3}$ of rainwater NO$_3^-$ and some of the river samples were calibrated with USGS-34 ($\Delta^{17}$O: -0.1 ‰ vs. VSMOW) and USGS-35 ($\delta^{18}$O +57.5 ‰ vs. VSMOW; $\Delta^{17}$O: +21.6 ‰ vs. VSMOW). The mass independent fractionation of NO$_3^-$ $^{17}$O vs. $^{18}$O ($\Delta^{17}$O vs. VSMOW) is calculated from Thiemens (1999):

$$\Delta^{17}O = \delta^{17}O - 0.52 \times \delta^{18}O \qquad (2)$$

The analytical reproducibility for $\Delta^{17}$O$_{NO3}$ averaged 0.3‰ based upon the pooled standard deviation of repeated measures of reference materials. The fraction (%) of atmospheric NO$_3^-$ in river water was derived from a two-end-member mixing equation of river water NO$_3^-$ ($\Delta^{17}$O = 0) with the corresponding atmospheric NO$_3^-$ $\Delta^{17}$O value (19.7 to 27.2 ‰; Section S1), with an associated uncertainty of ~1 % based on the pooled standard deviations of Monte Carlo error propagations.

### 2.6 *Particulate nitrogen analyses*

Particulate nitrogen (PN) in river samples was collected on pre-combusted 25 mm GF/F glass fiber filters that were freeze dried then compacted into tin capsules for analysis on a Costech Elemental Analyzer connected to a Thermo Delta V isotope ratio mass spectrometer via a Conflow IV interface. Samples were calibrated with aliquots of recognized reference materials USGS 40 and 41 ($\delta^{15}$N = -

4.52 and +47.57 ‰ vs. air, respectively), achieving an analytical precision of ~0.3 ‰.

### 2.7 *Nutrient flux estimates*

Instantaneous nutrient fluxes were estimated from the product of the nutrient concentration and the corresponding mean daily river discharge recorded by USGS gauges, or discharge reported by the Westerly WWTF. Given the large number of available river flow data from the USGS gauge at the

Stillman Bridge and effluent discharge from the WWTF in relation to comparatively fewer concentration data, annual fluxes of respective constituents were calculated using Beale's ratio estimator (Beale 1962; Quiblé et al. 2006), which accounts for the covariance between load and river flow values (Eq. 3):

$$L = \overline{CQ} \frac{\mu_q}{\overline{Q}} n \left( \frac{1 + \frac{1}{n_d} \frac{S_{CQ}}{\overline{CQ} \cdot \overline{Q}}}{1 + \frac{1}{n_d} \frac{S_{Q^2}}{\overline{Q}^2}} \right) \qquad (3a)$$

$$S_{CQ} = \frac{1}{n_d - 1} \left( \sum_{i=1}^{n_d} C_i Q_i - n_d \overline{Q} \, \overline{CQ} \right) \qquad (3b)$$

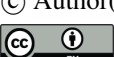



$$S_{Q^2} = \frac{1}{n_d - 1}\left(\sum_{i=1}^{n_d} Q_i - n_d \bar{Q}^2\right) \qquad (3c)$$

The term $\mu_q$ is the mean of all river discharge measurements, $C_i$ is the concentration on day i, $Q_i$ the average river discharge on day i, n the total number of days for the period of load estimation and $n_d$ is the number of observations of $C_i$. Overbars denote sample arithmetic means, and L is the resulting
load.

## 3.  Results

### 3.1 *Weekly river samplings*

The concentration of $NO_3^-$ measured in samples collected weekly at the Stillman Bridge was lowest in winter and highest in the summer months, ranging from to 9.7 µM to as high as 73.5 µM,
with a median value of 30.4 µM (Figure 2a). Comparable concentrations were detected at the Westerly Bridge at each sampling, except for instances where the site experienced saltwater intrusions, evidenced by elevated conductivities (data not shown) – at which times $[NO_3^-]$ at the Westerly Bridge was lower due to lower concentrations in the seawater endmember. The concentration of $NO_2^-$ was negligible in all samples. At both bridge sites, $[NO_3^-]$ decreased with
increasing river discharge (Figure 2b; Table 1). The $[NO_3^-]$ at the Stillman Bridge, upstream of potential seawater intrusion, also correlated directly with conductivity (Figure 3a). Values of $\delta^{15}N_{NO3}$ were lowest in winter and increased in summer, ranging from 5.3 ‰ to 9.4 ‰ – thus decreasing with increasing river discharge (Figure 2c-d; Table 1). Values of $\delta^{18}O_{NO3}$ followed a contrasting trend, being lower during the summer months and increasing in winter months, with values ranging from 1.6 ‰
to 6.8 ‰, barring a single an outlying value of 8.1 ‰ (Figure 2e). Values of $\delta^{18}O_{NO3}$ at the bridges increased directly with discharge (Figure 2f; Table 1). Measurements of $\Delta^{17}O_{NO3}$ at the Stillman Bridge ranged from -0.5 to 1.9 ‰. Uncycled atmospheric $NO_3^-$ was not detected in the majority of the river samples analyzed, with only 10 of 41 samples showing values above our lower limit of detection of ~1 % atmospheric $NO_3^-$. The fraction of atmospheric $NO_3^-$ was otherwise < 3%, notwithstanding a
single sample in which atmospheric $NO_3^-$ accounting for ~7 % of total riverine $NO_3^-$ (Figure 2g, S2; Section S1). Values of $\Delta^{17}O_{NO3}$ nevertheless correlated with river discharge (Figure 2h; Table 1).



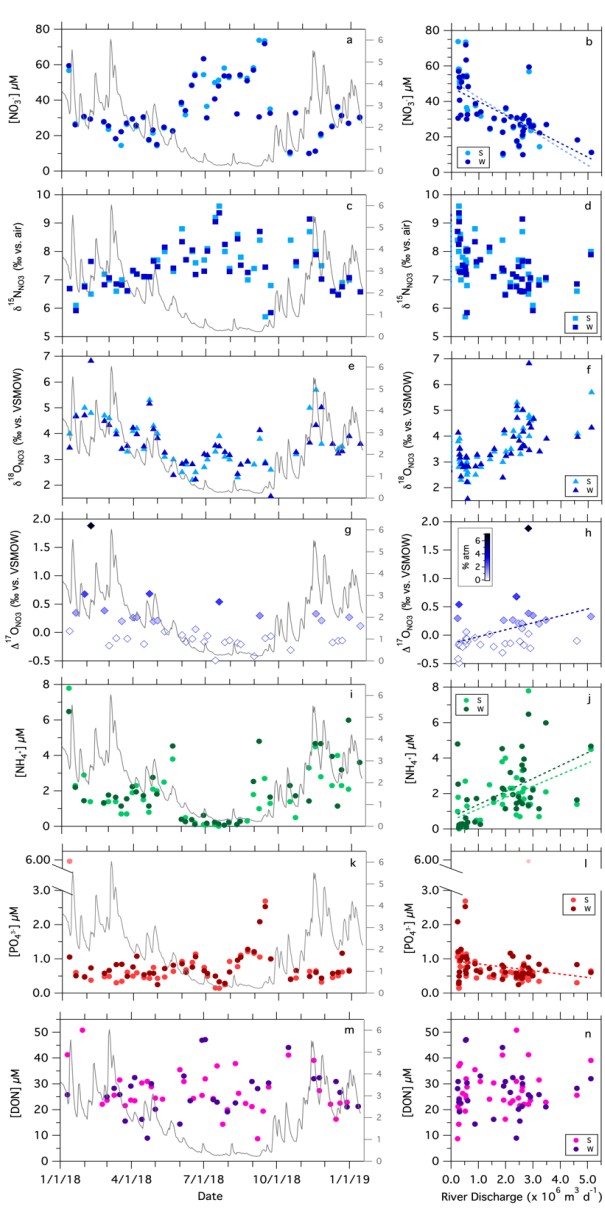

Figure 2. Weekly measurements of solute concentrations and $NO_3^-$ isotopologue ratios at the Stillman and Westerly bridges vs. the sampling date (superposed onto river discharge), and vs. the mean daily river discharge recorded at the Stillman Bridge. The secondary axis on left-hand panels is the river discharge (x $10^6$ $m^3$ $d^{-1}$). [$NO_3^-$] vs. (a) sampling date and (b) discharge; $\delta^{15}N_{NO3}$ vs. (c) sampling date and (d) discharge; $\delta^{18}O_{NO3}$ vs. (e) sampling date and (f) discharge; $\Delta^{17}O_{NO3}$ vs. (g) sampling date and (h) discharge; [$NH_4^+$] vs. (i) sampling date and (j) [$NH_4^+$] vs. discharge; [$PO_4^{3-}$] vs. (k) sampling date and (l) discharge; [DON] vs. (m) sampling date and (n) [DON] vs. discharge. Statistical fits of least-squares linear regressions are reported in Table 1.





The concentration of $NH_4^+$ recorded weekly at the bridges was consistently lower than corresponding [$NO_3^-$]. In contrast to [$NO_3^-$], [$NH_4^+$] at the bridges was lowest in summer and higher in winter, ranging

from below detection to 7.8 μM, and correlated directly with discharge (Figure 2i-j). The [$PO_4^{3-}$] ranged from 0.1 μM to 2.7 μM with one sample as high as 5.9 μM during a single sampling event,

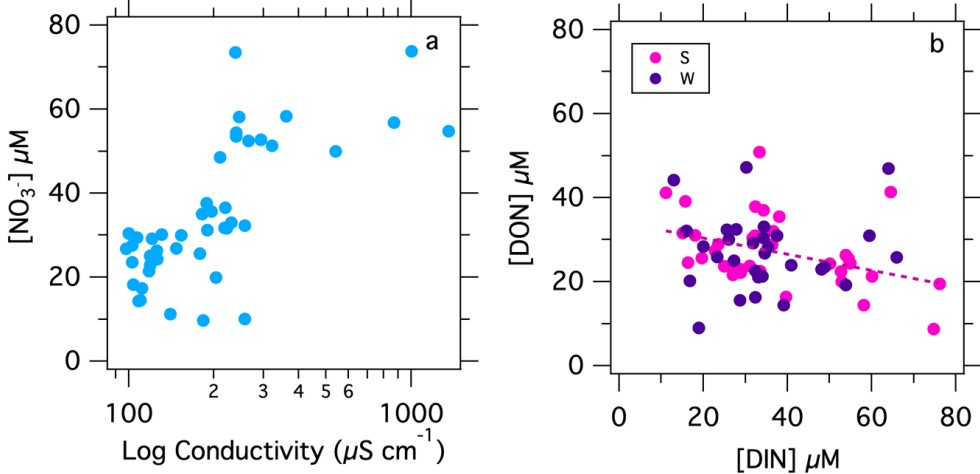

Figure 3. (a) [$NO_3^-$] in weekly samples at the Stillman Bridge vs. conductivity (log scale). (b) Weekly [DON] at the Stillman and Westerly bridges vs. [DIN].

exhibiting higher concentrations occurring in summer months, thus mirroring [$NO_3^-$] (Figure 2k). Concentrations of $PO_4^{3-}$ appeared to correlate inversely with discharge, yet only at the Westerly Bridge but not the Stillman Bridge (Figure 2l; Table 1).

The concentration of DON at the bridge sites ranged from 9 to 56 μM, appeared similar among seasons, and did not show a statistically significant relationship to river discharge (Figure 2m-n; Table 1). Nevertheless, [DON] and coincident [DIN] were inversely correlated, albeit weakly so, and significantly so only at the Stillman Bridge (Figure 3b; Table 1). In turn, [PN] exhibited median values of 2.6 μM from May through Oct, and 2.9 μM during the colder season, showing no values greater

than 7 μM; no correlation of [PN] with river discharge was evident (Figure S3a-b; Table 1). Concentrations of chlorophyll-a, which we measured only from June through December, ranged from 0.5 μg $L^{-1}$ to 12.1 μg $L^{-1}$, with higher values occurring in late summer to early fall. Chlorophyll-a showed no correlation with discharge (Figure S3e-f; Table 1).

    The daily riverine flux of dissolved inorganic nitrogen (DIN) delivered to the estuary from the

Pawcatuck River, computed from the product of river discharge and the sum of [$NO_3^-$] and [$NH_4^+$]



recorded at the bridges, varied ~10-fold over the annual sampling period, ranging from 0.1 to 1.1 (x $10^5$) moles of $N_{DIN}$ per day – omitting a single outlier of 1.8 x $10^5$ moles of $N_{DIN}$ per day (Figure 4a). The riverine DIN flux increased directly with river discharge, such that it was lowest in summer, averaging 0.2 ± 0.1 (x $10^5$) moles of $N_{DIN}$ per day from May through October (Figure 4b; Table 1). The

riverine DON flux, in turn, ranged from < 0.1 to 2.0 (x $10^5$) moles of $N_{DON}$ per day, and also increased directly with discharge (Figure 4c-d; Table 1). The total riverine N flux (TN flux), which is the sum of





respective DIN, DON and PN fluxes, ranged from 0.2 to 3.0 (x $10^5$) moles of $N_{TN}$ per day and correlated

directly with discharge (Figure 4e-f; Table 1).

### 3.2 *WWTF samples*

Nutrient concentrations measured in samples collected weekly at the Westerly WWTF, consisting

of both grab and composite samples, ranged from 30 to 527 µM for $[NO_3^-]$, 1.3 to 1070 µM for $[NH_4^+]$,

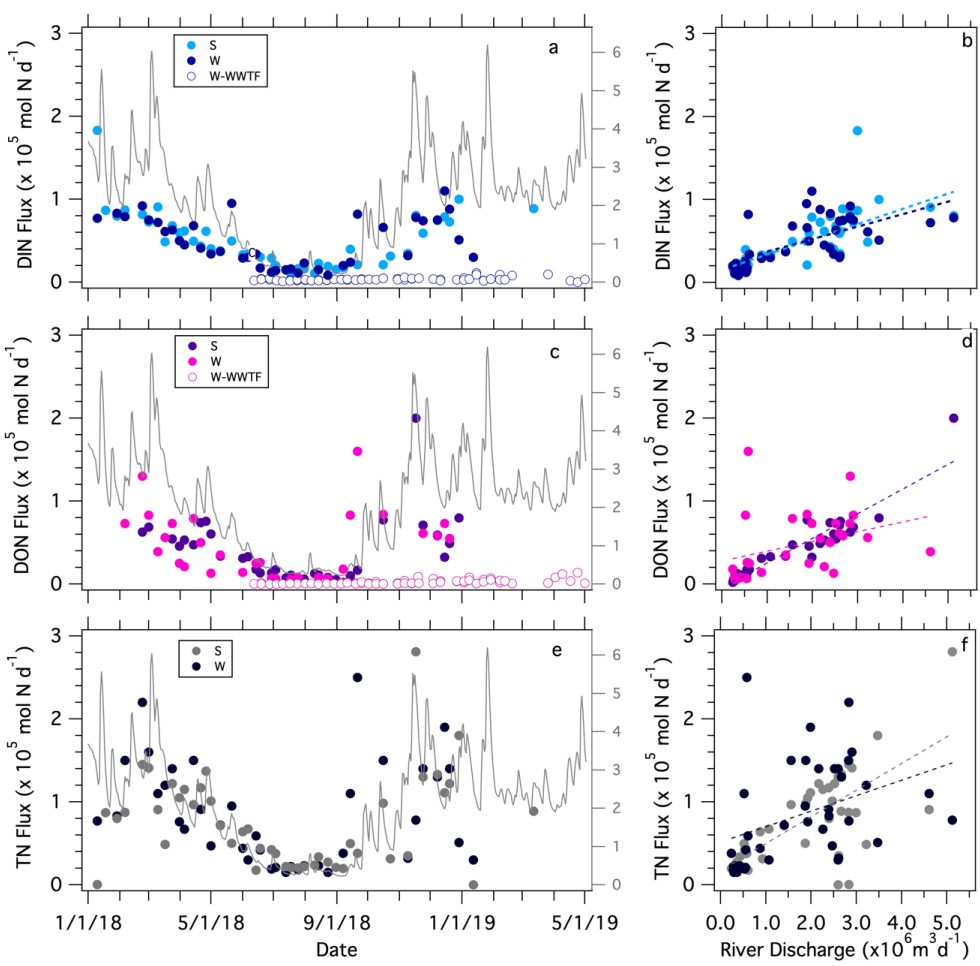

Figure 4. (a) DIN flux at the Stillman and Westerly bridges and from the Westerly WWTF vs. sampling
date; (b) DIN flux at the bridges vs. the mean daily river discharge recorded at the Stillman Bridge; (c)
DON flux at the bridges and from the W-WWTF vs. sampling date; (d) DON flux at the bridges vs. the
mean daily river discharge recorded at the Stillman Bridge; (e) TN flux at the bridges (the sum of DIN,
DON and PN) and from the W-WWTF vs. sampling date; (f) TN flux at the bridges vs. the mean daily
river discharge recorded at the Stillman Bridge.



11.7 to 1168 µM for [DON], and 2.7 to 26.5 µM for [$PO_4^{3-}$] (Figure 5a, c, e, g). Concentrations of $NO_3^-$ and $NH_4^+$ were highly correlated in grab vs. composite samples (Figure S4a-b). The [$NO_3^-$] and [$NH_4^+$] measured at UConn were similar to those reported by the Westerly WWTF (Figure S4c-d). The [DON]

measured at UConn showed poor correspondence to the facility-reported [TON] (total organic nitrogen) for the few corresponding sampling dates, although these sample types may arguably not be comparable as the UConn analyses did not include [PN] (Figure S4e).

Both [$NO_3^-$] and [$PO_4^{3-}$] were higher in summer months when facility discharge was lower, at which time [$NH_4^+$] was lower. The concentration of $NO_3^-$ correlated inversely with the facility-reported

discharge, whereas [$NH_4^+$] correlated directly with discharge (Table 1). There was an apparent increase in [DON] with discharge, albeit, with high variability during high flow in winter months, whereas facility-reported [TON] did not correlate with discharge (Figure 5f; Table 1). Our limited [$PO_4^{3-}$] measurements were not significantly correlated with facility-reported discharge (Figure 5hj; Table 1).

In contrast to the riverine N fluxes, which increased with river discharge, the DIN and TON fluxes from the Westerly WWTF were remarkably constant, and were substantially lower than corresponding riverine fluxes, averaging 3.2 x $10^3$ moles of $N_{DIN}$ per day, 1.0 x $10^3$ moles of $N_{TON}$ per day, and 4.1 x $10^3$ moles of $N_{TN}$ per day in 2018 (Figure 4 a-f; Table S1). The daily TN loading at the Westerly WWTF was notably lower than the permitted allowable daily discharge from May through

November of 13.5 x $10^4$ moles of $N_{TN}$ per day.





Table 1. Correlation coefficients, corresponding intercepts, coefficients of determination ($r^2$) and statistical probability of least-squared regression analyses from property-property plots of riverine solutes and fluxes. Statistically significant relationships are signaled by an asterisks (p-value ≤ 0.05*; ≤ 0.01**).


| X | Y | Location | Slope | Intercept | $r^2$ | p-value |
|---|---|---|---|---|---|---|
| River Discharge (x $10^6$ $m^3$ $d^{-1}$) | [$NO_3^-$] (µM) | S | -9.1 | 50.6 | 0.50 | ** |
| | | W | -7.9 | 47.8 | 0.44 | ** |
| | $\delta^{15}N_{NO3}$ (‰) | S | -0.3 | 7.9 | 0.16 | ** |
| | | W | -0.3 | 7.9 | 0.15 | ** |
| | $\delta^{18}O_{NO3}$ (‰) | S | 0.5 | 2.7 | 0.34 | ** |
| | | W | 0.4 | 2.9 | 0.31 | ** |
| | $\Delta^{17}O_{NO3}$ (‰) | S | 0.1 | -0.1 | 0.15 | * |
| | [$NH_4^+$] (µM) | S | 0.6 | 0.5 | 0.27 | ** |
| | | W | 0.7 | 0.6 | 0.29 | ** |
| | [$PO_4^{3-}$] (µM) | S | 0.0 | 0.8 | 0.00 | 0.78 |
| | | W | -0.1 | 1.0 | 0.09 | * |
| | [DON] (µM) | S | -0.2 | 34.2 | 0.15 | ** |
| | | W | -0.3 | 33.7 | 0.08 | 0.10 |
| | [PN] (µM) | S | 0.4 | 2.4 | 0.09 | 0.07 |
| | | W | 0.3 | 2.8 | 0.06 | 0.18 |
| | [chl-a] (µg $L^{-1}$) | S | -0.3 | 3.7 | 0.01 | 0.64 |
| | | W | -0.5 | 4.7 | 0.03 | 0.45 |
| | DIN flux (mol $d^{-1}$) | S | 2.6 x $10^4$ | Forced zero | 0.85 | ** |
| | | W | 2.7 x $10^4$ | Forced zero | 0.86 | ** |
| | DON flux (mol $d^{-1}$) | S | 3.0 x $10^4$ | Forced zero | 0.91 | ** |
| | | W | 2.7 x $10^4$ | Forced zero | 0.94 | ** |
| | TN flux (mol $d^{-1}$) | S | 6.0 x $10^4$ | Forced zero | 0.93 | ** |
| | | W | 6.0 x $10^4$ | Forced zero | 0.95 | ** |
| [DIN] (µM) | [DON] (µM) | S | -0.2 | 34.0 | 0.17 | ** |
| | | W | -0.3 | 33.7 | 0.08 | 0.10 |
| W-WWTF Discharge (x $10^4$ $m^3$ $d^{-1}$) | [$NO_3^-$] (µM) | WWTF comp | -268 | 514 | 0.43 | ** |
| | | WWTF comp-r | -240 | 439 | 0.31 | ** |
| | | WWTF grab | -268 | 544 | 0.42 | ** |
| | [$NH_4^+$] (µM) | WWTF comp | 232 | -83 | 0.09 | 0.10 |
| | | WWTF comp-r | 449 | -256 | 0.26 | ** |
| | | WWTF grab | 141 | -75 | 0.15 | * |
| | DON (µM) | WWTF comp | 262 | -8 | 0.10 | * |
| | | WWTF grab | 102 | 31 | 0.07 | 0.18 |
| | TON (µM) | WWTF comp-r | -11 | 115 | 0.00 | 0.75 |
| | [$PO_4^{3-}$] (µM) | WWTF comp | 5.0 x $10^{-2}$ | 8.0 | 0.05 | 0.30 |
| | | WWTF grab | 4.0 x $10^{-2}$ | 3.7 | 0.08 | 0.12 |
| 1/($NO_3^-$ flux) (d $mol^{-1}$) | $\delta^{15}N_{NO3}$ (‰) | S | 2.6 x $10^4$ | 6.6 | 0.43 | ** |
| | | W | 1.9 x $10^4$ | 6.8 | 0.39 | ** |
| | $\delta^{18}O_{NO3}$ (‰) | S | -2.1 x $10^4$ | 4.2 | 0.25 | ** |
| | | W | -1.0 x $10^4$ | 4.7 | 0.26 | ** |
| | $\delta^{18}O_{NO3}$-corr (‰) | S | -1.6 x $10^4$ | 3.8 | 0.26 | ** |



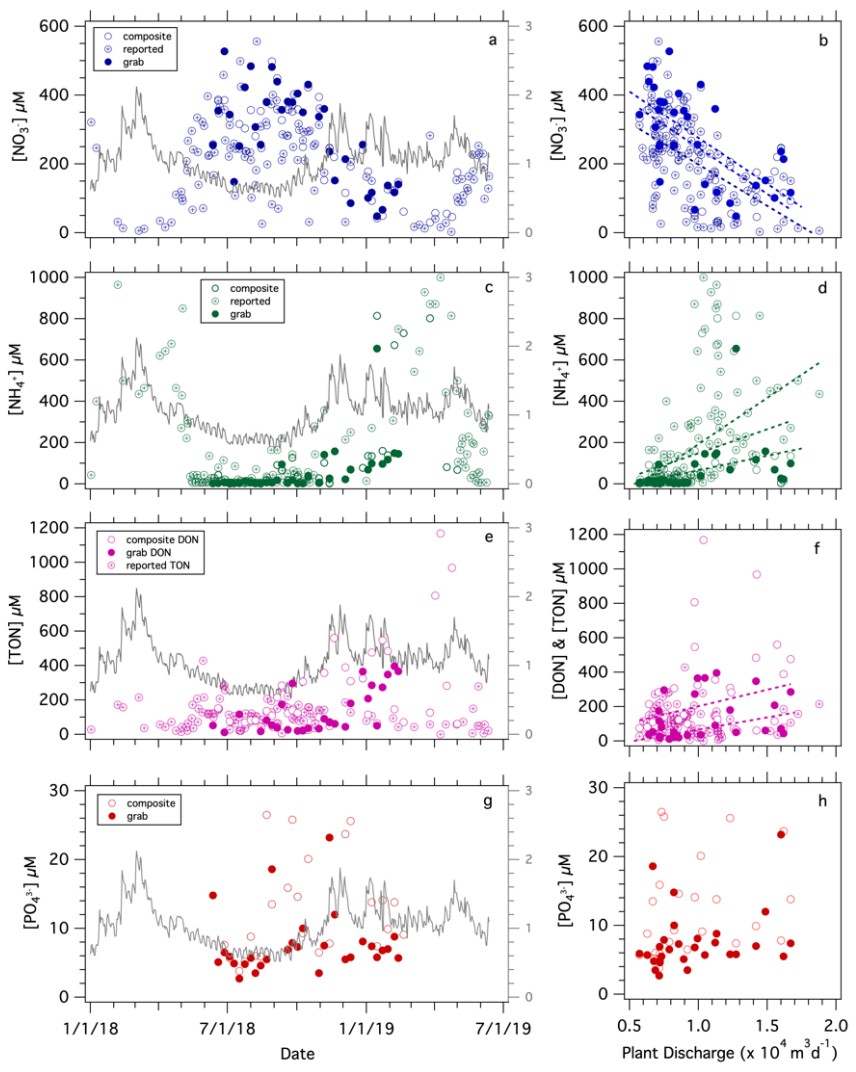

Figure 5. Nutrient concentrations discharged from the Westerly WWTF: [NO$_3^-$] vs. (a) date and (b) facility discharge; [NH$_4^+$] vs. (c) date and (d) facility discharge; [DON] and facility-reported [TON] vs. (e) date and (f) facility discharge; [PO$_4^{3-}$] vs. (g) date and (h) facility discharge. Grey line corresponds to the WWTF mean daily water discharge with reference to secondary axis on left-hand panels (x 10$^4$ m$^3$ d$^{-1}$). Statistical fits of least-squares linear regressions are reported in Table 1.

### 3.3 Along-river samplings

Samples collected at stations along the length of the river showed both spatial and seasonal patterns in nutrients and NO$_3^-$ isotope ratios (Figure 6, S5). On average, [NO$_3^-$] varied among sampling dates (F$_{2,12}$ = 122.4, p < 0.0001) and was lower during the November 2018 sampling than during the



May 2018 and March 2019 samplings at all river sites (Tukey HSD, both p < 0.05; Figure 6b; Table S2). [$NO_3^-$] tended to increase along river sections ($F_{3,9}$ = 32.1, p < 0.0001), but the specific patterns varied among sampling dates ($F_{6,12}$= 107.2, p < 0.0001). In the source basin at Worden Pond, [$NO_3^-$] ranged from 0.4 to 6 µM among sampling events and increased to date-specific maxima of 10 - 65 µM between Stations 2 and 4 (Biscuit City Road to Wood River Junction). Concentrations decreased

downstream of the Wood River inflow (between river sections 2 and 3) to values as low as 7 µM in November and as high as 29 µM in March (Tukey HSD, both p < 0.01), although these two sections of the river were similar in May 2018 (Tukey HSD, both p > 0.9). [$NO_3^-$] increased between Potter Hill Dam (Station 11) and the Stillman and Westerly bridges (between sections 3 and 4) during all sampling campaigns (Tukey HSD, all p < 0.05), with final concentrations of 10 µM in November and

32 µM in March (Figure 6b).

      Values of $\delta^{15}N_{NO3}$ differed among samplings dates ($F_{2,15}$ = 16.6, p < 0.001) and along the river ($F_{3,9}$ = 26.2, p < 0.001; Figure 6c). On average, values were lowest in March 2019, at which time [$NO_3^-$] was relatively elevated, than in May and November 2018 (Tukey HSD: both p < 0.001), although a sample in the uppermost river section (Station 2) had higher $\delta^{15}N_{NO3}$ values in March than in May (see

Discussion). Values during the March 2019 sampling ranged from 3.4 ‰ at Station 3 to 6.7 ‰ at the bridges (river section 4). Values in November 2018, which were similar to those in May 2018, ranged from 5.8 ‰ at Station 3 to 8.5 ‰ at the bridges. $NO_3^-$ delivered by the Wood River (Station 6) had $\delta^{15}N_{NO3}$ values similar to or greater than those of $NO_3^-$ originated upstream in the Pawcatuck River.

      In contrast to $\delta^{15}N_{NO3}$, $\delta^{18}O_{NO3}$ values tended to decrease downriver ($F_{3,9}$ = 8.6, p < 0.01; Figure

6d), despite relatively large variability. Relative maxima between 3.2 and 5.0 ‰ were apparent at Stations 3 and 4 (river section 2), decreasing to values to values oscillating between 2.7 and 4.8 ‰ toward the bridges ($F_{3,9}$ = 8.6, p < 0.01; Figure 6d). The $\delta^{18}O_{NO3}$ values upriver were generally higher in November (in contrast to $\delta^{15}N_{NO3}$) but otherwise occupied comparable ranges among sampling dates. Values contributed by the Wood River were similar to or marginally greater than those

upstream in the Pawcatuck River on corresponding dates.

      The concentration of $NH_4^+$ did not vary systematically across river sections ($F_{3,9}$ = 3.2, p = 0.08), ranging from 0.4 to 6.8 µM (Figures S5). [$NH_4^+$] was greater during the May 2018 sampling than during March or November, and this effect did not vary significantly across the river ($F_{6,12}$ = 2.1, p = 0.12).

      The concentration of $PO_4^{3-}$ varied over both space ($F_{3,9}$ = 45.2, p < 0.0001) and time ($F_{2,12}$ = 72.0,

p < 0.0001), and these effects were non-additive ($F_{6,12}$ = 32.2, p < 0.0001). [$PO_4^{3-}$] was relatively



homogeneous across the river sections in May 2018 (Tukey HSD, all pairwise p > 0.05) but increased in river section 2 in both March and November compared to neighboring sections up and downstream (Tukey HSD, all p < 0.01; Figure 6e).  Across all sampling dates, $[PO_4^{3-}]$ ranged from 0.2 to 0.5 µM in and near Worden Pond (river section 1) and peaked at values between 0.7 and 2.9 µM, in river section

2. Further downstream, $[PO_4^{3-}]$ ranged from 0.5 and 1.2 µM.  $[PO_4^{3-}]$ in the Wood River (Station 6) was relatively low and similar to that at Worden.

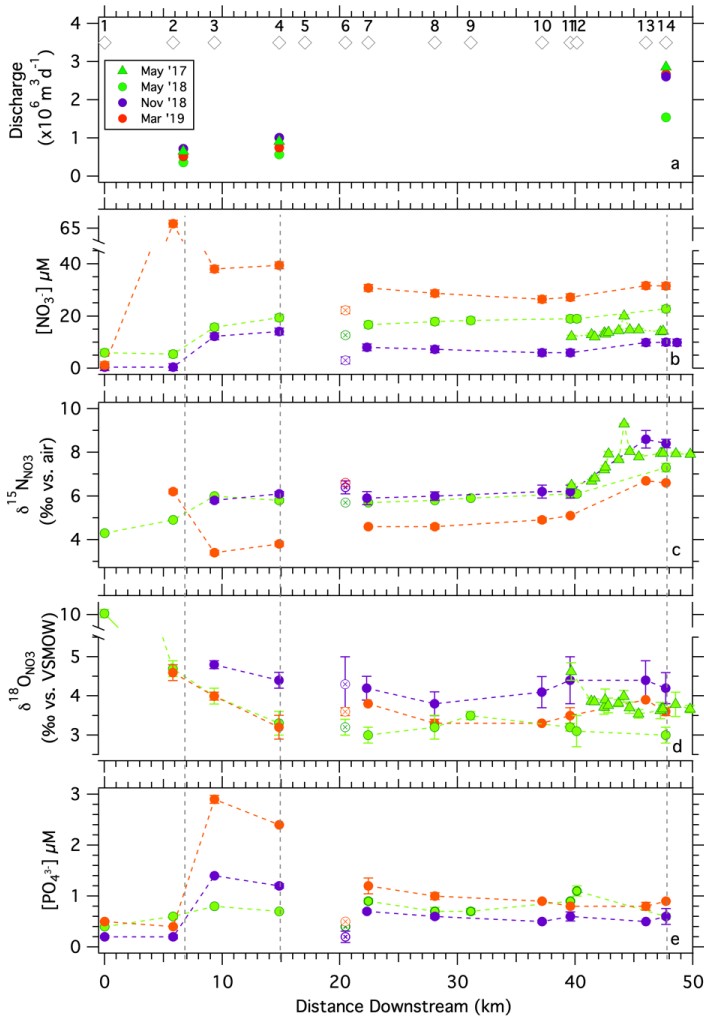

Figure 6. Solute concentrations observed at discrete locations along the Pawcatuck River from its origin at Worden Pond to the Westerly Bridge during along-river sampling campaigns. (a) Mean daily river discharge recorded at three flow gauges along the Pawcatuck River during the sampling campaigns; (b) $[NO_3^-]$, (c) $\delta^{15}N_{NO3}$, (d) $\delta^{18}O_{NO3}$, and (e) $[PO_4^{3-}]$ measured at stations along-river.





## 4. Discussion

### 4.1 *Nutrient source attribution*

At the Stillman and Westerly bridges, concentrations of $NO_3^-$ – the principal component of DIN –

scaled inversely with discharge, wherein higher concentrations occurred during summer at low base
flow. This relationship suggests the bulk of riverine DIN during low base flow originated from
groundwater and point sources along the river catchment. Given that there is only one documented
point source upstream of Stillman and Westerly bridges, we surmise that DIN at low base flow
originated predominantly from groundwater and partially from discharge at Kenyon Industries. That

$[NO_3^-]$ at the Stillman Bridge increased in proportion to conductivity also suggests a groundwater
source for bulk riverine nutrients at low base flow, although an analogous trend would admittedly
arise from loading by point sources.

During wetter months in winter, increased input of shallow groundwater and surface runoff
(henceforth collectively referred to as "shallow flow") diluted the low base flow $[NO_3^-]$, thus lowering

riverine concentrations, a dynamic documented in other rivers (Dubrovski et al. 2010). Nevertheless,
the daily DIN flux increased with discharge, indicating that DIN is also imported to the river by shallow
flow, albeit, at a lower concentration than low base flow DIN. From the slope of the DIN flux to
discharge relationship, the daily DIN flux increased by $2.6 \times 10^4$ moles of N per additional $10^6$ m$^3$ of
discharge, suggesting the DIN concentration of shallow flow from the catchment averaged 26 ± 3 µM

(Table 1). The relationship between [DIN] and river discharge, which we initially presumed linear, is
then better described by a two end-member mixing curve comprised of low base flow [DIN] mixing
with shallow flow [DIN] (Figure 7), assuming a 65 µM end-member approximated from the median
[DIN] at low base flow:

$$[DIN]_i = [(Q_i - 2.0 \times 10^8) * 26 \times 10^{-6} + 2.0 \times 10^8 * 65 \times 10^{-6}]/Q_i \qquad (4)$$

The term $Q_i$ is mean river discharge on day $i$ in units of L d$^{-1}$, and $[DIN]_i$ is the corresponding
concentration in units of moles L$^{-1}$. Implicit in Eq. 4 is the assumption of negligible in-river N
consumption, a notion supported by the low incident [PN]; we return to this dynamic further below.
The mixing relationship can serve to approximate the DIN flux from the Pawcatuck River into Little
Narragansett Bay based on the river discharge recorded continually by the USGS at the Stillman

Bridge.





The inverse correlation of [DON] with [DIN], in turn, suggests that [DON] is transported into the river by shallow ground water and surface flow from the catchment. Shallow flow, which increases with increased precipitation, is apt to transport organic material from soils and surface plant materials (Elwood and Turner, 1989; Mulholland et al. 1990; Pabich et al. 2001). The import of DON

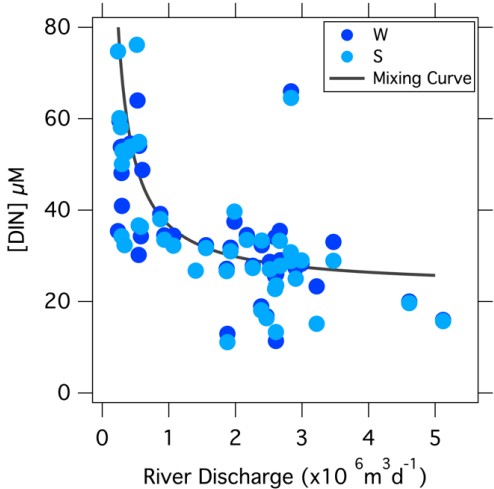

Figure 7: Mixing curve of low base flow [NO₃⁻] with shallow flow [NO₃⁻] superposed onto weekly measurements of [NO₃⁻] at the vs. the corresponding mean daily discharge at the Stillman Bridge (Eq. 4).

by shallow flow is consistent with the visibly elevated concentrations of riverine tannins. In this regard, the lack of direct correlation of [DON] to discharge is surprising, but may be masked by the relatively high variability of the [DON] measurements, even between replicate water samples.

Nutrient loading from the Pawcatuck River into Little Narragansett Bay was investigated previously by Fulweiler and Nixon (2005). As discerned herein, they observed an inverse relationship
of [DIN] to discharge from biweekly measurements at the Stillman Bridge over an annual cycle. Contrary to our interpretations, however, they argued that the decline in [DIN] with discharge was due to seasonal uptake by vegetation within the catchment, specifically during spring. They observed the lowest [DIN] in spring, corresponding to the highest discharge during their annual study period. Here, we otherwise argue that increased water discharge dilutes the low base-flow nutrients derived
from groundwater and point source discharge, such that concentrations are most elevated at low base flow. While the concentration is lower during periods of high river flow, the riverine DIN flux nevertheless increases with discharge, carrying nutrients imported by shallow flow.





Fulweiler and Nixon (2005) also observed that [DON] and [DIN] were inversely correlated, as in the current study, and further detected a positive correlation between [DON] and discharge, corroborating our earlier inference that such a relationship should be manifest. They reasoned that the greater remineralization of bioavailable DON in summer, at low discharge, could explain this trend, given the greater in-river residence time of DON at low base flow. While the mineralization of DON may be significant during the warm season (e.g., Brookshire et al. 2005), we otherwise contend that the increased [DON] with discharge may reflect import from the catchment via shallow flow.

The mean [DIN] imported by shallow flow inferred herein is relatively low (~26 µM), in the range of 15 to 70 µM generally observed in surface and shallow groundwater of undeveloped catchments across the US, and substantially lower than the range of 100 to 700 µM observed in shallow streams draining agriculture catchments (Dubrovsky et al., 2010). However, it is greater than the [DIN] of ≤ 5 µM recorded in shallow streams draining pristine forested catchments in the Northeast U.S.A., which are otherwise dominated by DON (Dickerman et al. 1989; Hedin et al. 1998). The [DIN] of ~65 µM recorded here at low base flow, which likely reflects that of deeper groundwater (barring a substantial point source input) is also within the range reported for groundwater $NO_3^-$ in undeveloped catchments, albeit, at the higher end of this range (of 7 to 75 µM; Mueller et al., 1995), and falling within the range reported for groundwater $NO_3^-$ in southern RI (0 - 91 µM; Moran et al., 2014). The mean low base flow [DIN] observed here is substantially lower than concentrations typical of groundwater in agricultural catchments, but higher than the [DIN] that was observed in the groundwater reservoir of the Upper Wood River in the 1980's (median ≤ 11 µM µM; Dickerman et al. 1989; Dickerman and Bell, 1993) – suggesting that anthropogenic input to the deeper groundwater N reservoir of the Pawcatuck watershed has increased over time.

### 4.2 Corroborating insights from $NO_3^-$ isotope ratios

We turn to the N and O isotope composition of $NO_3^-$ to further investigate relationships of nutrients with river discharge and to characterize N sources and cycling in the river. Like [$NO_3^-$], the isotope ratios of $NO_3^-$ co-varied with discharge. Values of $\delta^{15}N_{NO3}$ decreased with discharge, suggesting that (a) $NO_3^-$ added by shallow flow had lower $\delta^{15}N_{NO3}$ values than low base flow $NO_3^-$, and/or (b) $\delta^{15}N_{NO3}$ values at low base flow increased during warmer months compared to their groundwater end-member due to biological cycling in-river. Concurrently, $\delta^{18}O_{NO3}$ values increased with discharge, suggesting that (c) $NO_3^-$ added by shallow flow had higher $\delta^{18}O_{NO3}$ values than low



base flow $NO_3^-$, and/or (d) $\delta^{18}O_{NO3}$ values decreased in summer due to biological cycling. We consider these hypotheses in turn.

*4.2.1 Sources of DIN in shallow flow evidenced from $\delta^{15}N_{NO3}$ values*

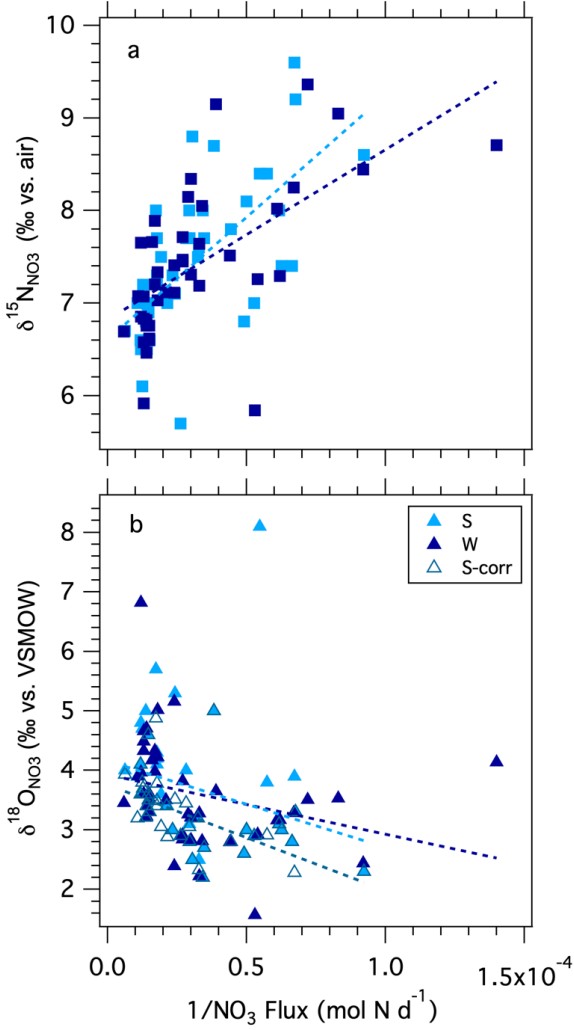

Figure 8. Modified Keeling Plot of $NO_3^-$ isotopologue ratios at the Westerly and Stillman bridges vs. the inverse of the daily $NO_3^-$ flux: (a) $\delta^{15}N_{NO3}$ vs. the inverse of the $NO_3^-$ flux, (b) $\delta^{18}O_{NO3}$ and $\delta^{18}O_{NO3}$ corrected for atmospheric $NO_3^-$ vs. the inverse of the $NO_3^-$ flux.

In order to evaluate whether the lower $\delta^{15}N$ DIN values observed at higher discharge can be explained by the addition of relatively low $\delta^{15}N$ DIN by shallow flow, we plotted the $\delta^{15}N_{NO3}$ values





recorded at the Stillman Bridge vs. the inverse of the corresponding $NO_3^-$ flux (i.e., an adapted Keeling

Plot; Keeling, 1958, 1961; Figure 8a). Because we lack measurements of the $\delta^{15}N$ values of the

incident $NH_4^+$ pool (which we could not assess due to an analytical interference from dissolved

organic material; *see* Zhang et al. 2007), we assume that the N isotope composition of $NO_3^-$ captures

that of bulk DIN, on the basis that $NH_4^+$ imported from the catchment was largely nitrified in-river,

wherein $NH_4^+$ accounted for only a small fraction of the DIN reservoir. The riverine $\delta^{15}N_{NO3}$ data

conform to a linear relationship expected for the addition of DIN with a lower mean $\delta^{15}N$ to the low

base flow reservoir (Table 1). The intercept of the resulting linear regression suggests that the $NO_3^-$

associated with increased discharge had a mean $\delta^{15}N$ value of ~6.7 ‰ (Table 1), compared to a low

base flow value of ~8 ‰ observed at the bridges. The average $\delta^{15}N_{NO3}$ value of atmospheric $NO_3^-$ in

rainwater was -2.5 ± 2.1 ‰ (Section S1; Figure S2), indicating that $NO_3^-$ added by shallow flow did not

originate predominantly from direct atmospheric deposition as uncycled atmospheric $NO_3^-$. While the

$\delta^{15}N_{NO3}$ of atmospheric $NO_3^-$ could conceivably be fractionated by biological cycling in-river following

its import by shallow flow, increased discharge occurred largely during the cold season, at which time

biological cycling in-river was presumably curtailed. Thus, we surmise that the DIN added by shallow

flow did not originate from direct atmospheric deposition as uncycled atmospheric $NO_3^-$, but rather

derived from catchment soils and shallow groundwater. The $\delta^{15}N_{NO3}$ end-member value of ~6.7 ‰ is

in the upper range observed for soil $NO_3^-$ in temperate forested catchments (Mayer et al. 2002;

Barnes and Raymond, 2009). While the net sources of reactive N to forested soils are atmospheric

deposition and biological $N_2$ fixation – which have relatively low $\delta^{15}N$ values (≤ 0‰) – partial

denitrification in soils and shallow groundwater increases the $\delta^{15}N$ of the soil N reservoir to values of

~5 ‰ (Amundson et al., 2003; Houlton et al., 2006; McMahon and Böhlke 2006; Houlton and Bai,

2009). The $NO_3^-$ imported by shallow flow draining urbanized systems has comparatively higher

$\delta^{15}N_{NO3}$ values (≥ 10 ‰; e.g., Divers et al., 2014), while $NO_3^-$ in soils and shallow groundwater in

agricultural systems generally falls within a lower range of values between 2 - 4 ‰ (Green et al. 2008;

Böhlke et al. 2009; Lin et al., 2019). The watershed of the Pawcatuck River is largely forested, yet

hosts agricultural and urbanized sections that ostensibly contributed to the mean $\delta^{15}N_{NO3}$ end-

member imported by shallow flow. Thus, while DIN added to the river by shallow flow at high

discharge had a mean $\delta^{15}N$ value consistent with expectations for a largely forested catchment, inputs





from agricultural and urbanized catchments may be rendered undiscernible due to their opposing contributions to the mean $\delta^{15}N_{NO3}$ value in shallow flow.

4.2.2 *Negligible fraction of uncycled atmospheric $NO_3^-$ confirmed by O isotope ratios*

The inference that uncycled atmospheric $NO_3^-$ did not contribute substantially to the increased $NO_3^-$ flux at higher discharge is corroborated by the $\Delta^{17}O_{NO3}$ measurements at the Stillman Bridge. The low values observed evidenced only a slight contribution of < 3% uncycled atmospheric $NO_3^-$ to total riverine $NO_3^-$ in a few samples, suggesting efficient processing of atmospheric $NO_3^-$ in soils

shallow groundwater (Mengis et al, 2001; Barnes et al. 2008). This observation is further echoed in a recent metanalysis of North American rivers, wherein the contribution of uncycled atmospheric $NO_3^-$ to base flow was inferred to be generally modest (Sebestyen et al. 2019). The $NO_3^-$ delivered to the Pawcatuck River by shallow flow evidently originated from a reservoir that was biologically cycled within catchment soils – and potentially in-river – thus losing its atmospheric $\Delta^{17}O$ signature.

A Keeling plot of $\delta^{18}O_{NO3}$ values vs. the inverse of the $NO_3^-$ flux at the bridges suggests that $NO_3^-$ added by surface flow had a mean $\delta^{18}O_{NO3}$ value of ~4.5 ‰ (Figure 8b; Table 1), compared to a mean low base flow value of 2.8 ± 0.2 ‰. Although the contribution of uncycled atmospheric $NO_3^-$ to the riverine reservoir was modest, we nevertheless consider that the increase in $\delta^{18}O_{NO3}$ values with discharge may derive in part from uncycled atmospheric $NO_3^-$, given the direct relationship of $\Delta^{17}O_{NO3}$

to discharge, and considering the characteristically elevated $\delta^{18}O_{NO3}$ values of 60 - 80 ‰ observed in the local rainwater $NO_3^-$. Indeed, when the weighted contribution of atmospheric $NO_3^-$ is subtracted from individual $\delta^{18}O_{NO3}$ values (attributed from corresponding $\Delta^{17}O$ measurements, accounting for precipitation-dependent differences in the mean $\Delta^{17}O$ and $\delta^{18}O_{NO3}$ values of rainwater), the intercept of the Keeling plot decreases slightly to ~3.8 ‰, nevertheless remaining greater than the $\delta^{18}O_{NO3}$ of

low base flow $NO_3^-$ (Table 1). The higher $\delta^{18}O_{NO3}$ with higher discharge is thus partially explained by the small component of uncycled atmospheric [$NO_3^-$] with elevated $\delta^{18}O_{NO3}$ values.

The $\delta^{18}O_{NO3}$ signature of 3.8 ‰ for $NO_3^-$ added with increasing discharge (minus the uncycled atmospheric $NO_3^-$) is in the range generically observed for soil $NO_3^-$ (Kendall et al., 2007; Michener and Lajtha, 2007). It has traditionally been ascribed to that expected for newly nitrified $NO_3^-$, based

on an empirical metric stipulating that the $\delta^{18}O_{NO3}$ values produced by nitrification derive from the fractional contribution of the reactants, namely 1/3 $\delta^{18}O$ of $O_2$ + 2/3 $\delta^{18}O$ of $H_2O$ (Anderson and Hooper, 1983; Hollocher 1984; Kendal et al., 2007). Considering that the $\delta^{18}O_{H2O}$ of Pawcatuck river





water is -7 ‰ and the $\delta^{18}O_{O2}$ of atmospheric oxygen is ~23.5 ‰ (Kroopnick and Craig, 1972), the nitrification $\delta^{18}O_{NO3}$ value thus expected is on the fortuitous order of 3.2 ‰. This empirical metric,

however, demonstrably overlooks substantive isotope effects associated with O-atom incorporation into the $NO_3^-$ molecule during nitrification and isotopic exchange of the nitrite intermediate with water, which otherwise give way to nitrified $NO_3^-$ whose $\delta^{18}O_{NO3}$ value is close to that of ambient water (Sigman et al., 2009; Casciotti et al., 2008; Buchwald and Casciotti, 2010; Snider et al., 2010; Boshers et al., 2019). This consideration explains frequent observations of relatively low $\delta^{18}O_{NO3}$ in

some soils and saturated systems, which are not explained by simple fractional contribution of reactants (Hinkle et al. 2008; Xue et al., 2009; Fang et al. 2012; Veale et al., 2019). Thus, we posit that the O isotope composition of the $NO_3^-$ imported into the river with increased discharge, which is typical of that in soils and shallow groundwater, does not strictly indicate that shallow flow $NO_3^-$ originated from proximate nitrification therein, as generally presumed, but also signals that $NO_3^-$

underwent partial denitrification in soils and shallow groundwater, resulting in a coupled increase in its $\delta^{15}N$ and $\delta^{18}O$ relative to source values (Houlton et al. 2006; Granger and Wankel 2016; Boshers et al. 2019). Although increased discharge occurred largely in winter, some in-river biological cycling during colder months could additionally influence the shallow flow $\delta^{18}O_{NO3}$ end-member, specifically reducing it from its soil value due to the nitrification of incident $NH_4^+$. Thus, $\delta^{18}O_{NO3}$ values imported

by shallow flow, once adjusted for modest contributions of uncycled atmospheric $NO_3^-$, fall within the range typically observed in soils, potentially modified by nitrification in-river.

### 4.2.3 *Values of $\delta^{15}N_{NO3}$ at low base flow reflect groundwater DIN*

The higher $\delta^{15}N_{NO3}$ values at low base flow compared to shallow flow may derive directly from those of the ground-water end-members and point source(s); The $\delta^{15}N_{NO3}$ values in deeper

groundwater are generally higher than in shallower groundwater above, fractionated by denitrification in the saturated zone. Alternatively, the higher $NO_3^-$ isotope ratios at low base flow may result from increased biological cycling in summer – modifying the isotope composition of low base flow $NO_3^-$ relative to its groundwater and/or point source values. The expectation of increased biological activity in summer months is consistent with the incident decrease in $[NH_4^+]$ with lower

discharge, which can be explained by a seasonal increase in algal assimilation and nitrification. Fulweiler and Nixon (2005) similarly observed lower $[NH_4^+]$ in the summer, but saw no correlation to



river discharge, further supporting our contention that increased seasonal biological cycling underlies the [$NH_4^+$] dynamics, rather than river discharge.

The extent to which the coincident $NO_3^-$ pool is also assimilated during summer months – and isotopically fractionated – is unclear. The fraction of the $NO_3^-$ pool assimilated by algae may be modest, even in summer, on the basis that the phytoplankton biomass was relatively small due to the high tannin content of the river water, which limits light penetration. Median chlorophyll-a concentrations in summer were ~1.3 µg L$^{-1}$ at the Stillman and Westerly bridges – save for late summer where higher concentrations were detected – while the median [PN] was ~2.5 µM, and no

greater than 7 µM. There are, however, populations of emergent plants along some shallow reaches of the river, which may assimilate $NO_3^-$ as well as reduced N substrates. Nevertheless, the inference that the riverine $NO_3^-$ pool is minimally assimilated, even in summer, appears consistent with along-river distribution of $NO_3^-$ isotope ratios. If a sizeable fraction of the incident $NO_3^-$ pool were assimilated into biomass during summer months, both the $\delta^{15}N_{NO3}$ and $\delta^{18}O_{NO3}$ values of low base

flow $NO_3^-$ would expectedly increase in proportion to the fraction of $NO_3^-$ assimilated (Granger et al., 2004; Johannsen et al., 2008). However, the $\delta^{15}N_{NO3}$ increase along-river observed during the seasonal surveys, which could be construed as signaling partial assimilation of riverine $NO_3^-$, was not matched by coincident along-river increases in $\delta^{18}O_{NO3}$ values. Similarly, [PN] and chlorophyll-a did not increase along-river, as would otherwise be expected for the progressive and sizeable conversion

of the $NO_3^-$ pool into the particulate pools (Figure S6c-d). Thus, we rule out a dominant influence of algal assimilation in fractionating the riverine $NO_3^-$ isotope ratios.

A more nuanced framework from which to interpret the $NO_3^-$ isotope ratios is afforded by the concept of riverine nutrient spiraling, namely, the continual assimilation of nutrients in the water column, the remineralization of organic material in sediments, and the return of remineralized

nutrients to the water column where they can undergo assimilation into new biomass (*reviewed by* Ensign and Doyle, 2006; Harvey et al., 2013). A small fraction of the $NO_3^-$ pool is likely assimilated during the growing season, resulting in the production of PN with a lower $\delta^{15}N$ than coincident $NO_3^-$ due to N isotope fractionation during assimilation (Needoba et al. 2003; Figure 9). Considering the small summertime pools of PN and $NH_4^+$ relative to the $NO_3^-$ pool, $\delta^{15}N_{NO3}$ values will be minimally

fractionated by assimilation. Moreover, the concomitant recycling of PN and its subsequent nitrification will ostensibly regenerate $NO_3^-$ with a $\delta^{15}N_{NO3}$ value roughly equivalent to that assimilated into organic material then ammonified – given an approximate steady state between $NO_3^-$



assimilation and nitrification – such that $\delta^{15}N_{NO3}$ values will not incur a progressive increase from continual assimilation along-river. These dynamics will result in little net change in riverine $\delta^{15}N_{NO3}$

values relative to the mean catchment end-member.

The $NO_3^-$ isotope ratios could, however, be influenced by denitrification in-river (Kellman and Hillaire-Marcel 1998; Figure 9). While direct benthic denitrification does not communicate an isotope enrichment to $NO_3^-$ in the overlying water column due to a reservoir effect (Brandes and Devol, 1997; Sebilo et al., 2003; Lehmann et al., 2005), $\delta^{15}N$- and $\delta^{18}O$-enriched $NO_3^-$ from the sediment depth of

denitrification can be entrained into the water column by hyporheic flows in the riparian zone (Sebilo et al., 2003). Moreover, coupled nitrification-denitrification can fractionate the N isotopologues of $NH_4^+$ in surface sediments in proportion to the corresponding fraction of nitrified $NO_3^-$ lost concurrently to denitrification, thus contributing to an increase in $\delta^{15}N$ of the water column reactive N reservoir (Brandes and Devol 1997; Granger et al, 2011).

The along-river increase in $\delta^{15}N_{NO3}$ values could then result from isotopic fractionation by sedimentary denitrification. Yet a downstream increase in $\delta^{15}N_{NO3}$ was notably apparent in all seasons, not only in summer. On the presumption that water-column and benthic N cycling were substantially reduced during the March 2019 sampling when river waters were colder (average temperature of 5.9 °C), we surmise that the increase in $\delta^{15}N_{NO3}$ values along-river arises principally from differences in

the $\delta^{15}N$ of DIN input from respective reaches of the catchment – although some influence of benthic denitrification on riverine $\delta^{15}N_{NO3}$ values cannot be ruled out. We thus interpret the riverine $\delta^{15}N_{NO3}$ values to predominantly reflect the N isotope composition of DIN input from the catchment. We return to this insight in a subsequent section, to identify N sources along the catchment.

4.2.4 *Influence of in-river biological cycling on $\delta^{18}O_{NO3}$ values at low base flow*

The $\delta^{18}O_{NO3}$ values along-river can also be interpreted within the framework of nutrient spiraling. As with $\delta^{15}N_{NO3}$, the riverine $\delta^{18}O_{NO3}$ values integrate the contribution of $NO_3^-$ imported from the catchment (including uncycled atmospheric $NO_3^-$), the $NO_3^-$ produced by nitrification in-river, and the $NO_3^-$ consumed by assimilation and by denitrification (Figure 9). Without continual exogenous input from the catchment, $\delta^{18}O_{NO3}$ values of an initial $NO_3^-$ reservoir would theoretically converge

downriver onto a steady-state value dictated by the $\delta^{18}O_{NO3}$ of newly nitrified $NO_3^-$ and the effective isotope effect for $NO_3^-$ consumption, by assimilation and denitrification: For instance, assuming a $\delta^{18}O_{NO3}$ value of -6 ‰ for newly nitrified $NO_3^-$ ($\delta^{18}O_{H2O}$ + 1 ‰; Casciotti et al., 2008; Sigman et al.,



2009; Buchwald & Casciotti, 2010, Granger et al., 2013; Boshers et al., 2019), a canonical $NO_3^-$ assimilation isotope effect of 5 ‰ (Needoba et al., 2003), and no influence of sedimentary

denitrification on water column $\delta^{18}O_{NO3}$, values downriver would asymptote to -1 ‰. The $\delta^{18}O_{NO3}$ values of ~3 ‰ observed at the bridges during low base flow thus suggest that the $NO_3^-$ introduced continuously along the catchment had $\delta^{18}O_{NO3}$ values greater than -1 ‰ – assuming roughly equivalent in-river assimilation and nitrification fluxes. These greater $\delta^{18}O_{NO3}$ values may also signal some influence of sedimentary denitrification in fractionating the water-column $\delta^{18}O_{NO3}$.

Observations of decreasing along-river values are then consistent with the notion of higher catchment $\delta^{18}O_{NO3}$ end-member values converging onto lower values determined by the ratio of nitrification to consumption in-river – and associated isotopic fractionation. Within this framework, $\delta^{18}O_{NO3}$ values in winter, when biological cycling is dampened, would expectedly increase to values closer to the catchment sources, a prediction that appears to be borne out in our observations.

Barnes et al. (2008) similarly observed higher $\delta^{18}O_{NO3}$ values during the cold season in steams draining forested watersheds in the Northeastern U.S.A. The riverine $\delta^{18}O_{NO3}$ values thus afford insights into N sources and cycling that are consistent with expectations for nutrient spiraling.



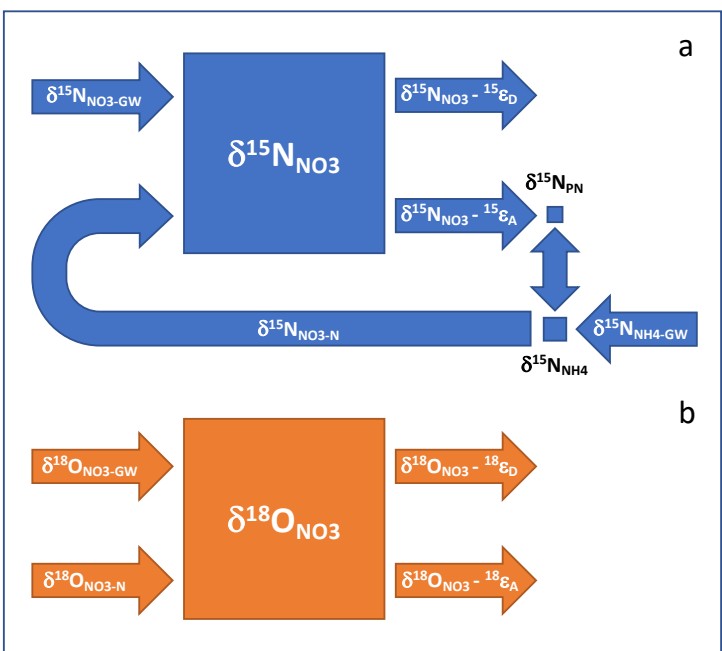

Figure 9. Conceptual illustration of the influence of nutrient spiraling on the N and O isotope ratios of riverine $NO_3^-$. Nutrient spiraling describes the cycling of nutrients as they are assimilated from the water column into biomass that is temporarily retained on the benthos, then mineralized and released back into the water column or denitrified. (a) The $\delta^{15}N$ of the riverine $NO_3^-$ reservoir integrates the $NO_3^-$ and $NH_4^+$ delivered continually from groundwater ($\delta^{15}N_{NO3-GW}$ and $\delta^{15}N_{NH4-GW}$), minus the $NO_3^-$ removed concurrently by sedimentary denitrification – the $\delta^{15}N$ of which depends on the sedimentary isotope fractionation communicated to the water column reservoir, $^{15}\varepsilon_D$. Given the small size of the respective PN and $NH_4^+$ pools relative to $NO_3^-$, ammonification and subsequent nitrification produce $NO_3^-$ with a $\delta^{15}N_{NO3-N}$ value approximating that lost concurrently to assimilation ($\delta^{15}N_{NO3} - ^{15}\varepsilon_A$), notwithstanding the $NH_4^+$ input from groundwater. The input of groundwater $NH_4^+$ ($\delta^{15}N_{NH4-GW}$) implicitly subsumes the input of reactive allochthonous PN and DON. (b) The riverine $\delta^{18}O_{NO3}$ integrates the $NO_3^-$ input from groundwater and precipitation ($\delta^{18}O_{NO3-GW}$) and from in-river nitrification ($\delta^{18}O_{NO3-N}$), minus $NO_3^-$ lost to algal assimilation and sedimentary denitrification – whose respective values depend on the net isotope effects associated with assimilation and denitrification, $^{18}\varepsilon_A$ and $^{18}\varepsilon_D$.

### 4.2.5 Regional N sources to the Pawcatuck River

Observations from the along-river surveys provide insights into the contribution of different reaches of the catchment to the riverine N reservoir. Areas of disproportionate loading can be identified from distinct concentration increases, and areas of lesser loading and/or net attenuation from concentration decreases. Reaches of the river that exhibit disproportionate loading present potential targets for mitigation. As detailed above, we interpret changes in $\delta^{15}N_{NO3}$ values along-river

to reflect differences in the $\delta^{15}N$ of DIN inputs from respective reaches of the catchment, thus serving to identify dominant regional N sources.



Surface water in Warden Pond had relatively low nutrient concentrations, which remained similarly low at Biscuit City Road (Station 2) in two of three samplings. The otherwise extremely elevated [NO$_3^-$] at Station 2 in March 2018 decreased downstream at Station 3 by > 40 %, more than

can be explained by either dilution from additional inflow or by denitrification. This elevated concentration may then reflect the inadvertent sampling of a groundwater plume or a localized reach of slow-flowing water, rather than the mean river composition. Monitoring at Biscuit City Road from 2007-2016 by the Wood-Pawcatuck Watershed Association similarly reveals relatively low median [NO$_3^-$] values of ~1 µM during fall samplings, punctuated by stochastic instances of elevated

concentrations, as high as 41 µM (Figure S6; Wood-Pawcatuck Watershed Association, 2020). The $\delta^{15}N_{NO3}$ value recorded at this station during the March 2019 sampling was 6.2 ‰ and the $\delta^{18}O_{NO3}$ was 4.8 ‰, values consistent with either a groundwater plume or a slow-flowing reach of the river.

Both [NO$_3^-$] and [PO$_4^-$] increased thereon at Kenyon and Wood River Junction (Stations 3 and 4, respectively) in all sampling campaigns. Associated $\delta^{15}N_{NO3}$ values were relatively low during the

March 2019 sampling (≤ 4 ‰) – coincident with more elevated [NO$_3^-$] – potentially signaling the input of DIN by shallow flow from proximate turf farms (Kreitler et al., 1978 ; Katz et al., 1999; Townsend et al., 2002; Deutsch et al, 2005). Input of uncycled atmospheric NO$_3^-$ by surface flow due to reginal snow melt, which could also explain lower $\delta^{15}N_{NO3}$ values, is not supported by the corresponding $\delta^{18}O_{NO3}$ values, which would otherwise be disproportionately high. Moreover, there was little to no

snow accumulated in March 2019. The increased nutrient concentrations observed at Stations 3 and 4 in all sampling campaigns also likely derived in part from the retention ponds at Kenyon Industries, in light of a permitted discharge of 7,500 moles N and 950 moles P per day (U.S. Environmental Protection Agency, 2010). Corresponding $\delta^{15}N_{NO3}$ values at Stations 3 and 4 during the May and November 2018 samplings were ~6 ‰, which could indicate input from deeper agricultural

groundwater, or could reflect discharge by Kenyon Industries, for which we do not have end-member values.

Inflow from the less impacted Wood River evidently diluted nutrient concentrations in the Pawcatuck River (Station 7). The Wood River contributes significantly to the total discharge of the Pawcatuck river (≥ 14 ± 5 % of total – based on discharge at Hope Valley USGS gauge), draining a more

forested watershed that harbors fewer agricultural areas than the lower Pawcatuck River. The [NO$_3^-$] in all sampling campaigns remained relatively invariant downstream of the Wood River inflow through the largely forested catchment to Potter Hill Dam (Station 11), while $\delta^{15}N_{NO3}$ values increased




marginally. The increases in [NO$_3^-$] and $\delta^{15}$N$_{NO3}$ thereon to the Stillman and Westerly bridges indicate DIN input from groundwater in the more populated portion of the watershed. The population density and associated septic systems increase considerably in the vicinity of the town of Westerly (Wood-Pawcatuck Watershed Association, 2016). Septic leachate and urban runoff are typically associated with relatively higher $\delta^{15}$N values, on the order of 8 to 15 ‰ (Kendall et al. 2007; Böhlke et al., 2009; Kasper et al, 2015). Thus, changes in land use along the catchment best explain the $\delta^{15}$N$_{NO3}$ increase in the lower portion of the river.

In all, the substantial difference in [DIN] between Stations 2 and 5 signals disproportionate input from this section of the watershed, likely owing to the proximity of turf farms and discharge from Kenyon Industries. Indeed, the riverine DIN flux at Wood River Junction amounted to 28 ± 11 % of the DIN flux recorded at the Stillman bridge among the 3 sampling dates, while accounting for only 11 ± 2 % of the riverine discharge. A fraction of the N loaded in this portion of the river may arguably be partially attenuated by denitrification along-river; nevertheless, this regional input remains substantial even assuming some biological attenuation. This portion of the river also contributed disproportionately to the riverine PO$_4^{3-}$ burden, although we do not explicitly consider this contribution in relation to the total discharge into the estuary, given the complex geochemistry of PO$_4^{3-}$ that involves adsorption and release from authigenic particles in sediments (Froelich, 1988).

The increase in [DIN] and $\delta^{15}$N$_{NO3}$ values in the lower portion of the river, in light of the large coincident river discharge, also signals a disproportionate contribution from the urbanized portion of the catchment. However, lacking estimates of river discharge at Potter Hill Dam (Station 11), we cannot deduce the fractional contribution from this portion of the watershed confidently. Nevertheless, assuming a $\delta^{15}$N input from the urbanized catchment of 10 ‰ and a mean $\delta^{15}$N$_{NO3}$ of 6‰ at Potter Hill Dam, compared to 8 ‰ at the Westerly Bridge, the DIN added to the river within this reach would amount to ~50 % of the total riverine N load. Otherwise assuming a $\delta^{15}$N input of 15 ‰, the DIN contributed from the urbanized reach would otherwise amount to ~20 % of total.



### 4.3 *N loading into Little Narragansett Bay*

#### 4.3.1 *Riverine contributions*

Estimates of the annual N loading from the Pawcatuck River into Little Narragansett Bay for 2018, compiled from our weekly measurements at the Stillman bridge, were $20.2 \times 10^6$ moles yr$^{-1}$ for DIN and $40.3 \times 10^6$ moles yr$^{-1}$ for TN, albeit, with uncertainty associated with the TN loading estimate given the variability of our DON measurements (Table 2). These values are considerably larger than

those estimated from biweekly measurements at the Stillman Bridge for 2002 by Fulweiler and Nixon (2005), which were $7.2 \times 10^6$ moles yr$^{-1}$ for DIN and $16.0 \times 10^6$ moles yr$^{-1}$ for TN. The greater N loading in 2018 could arise from (a) an increase in groundwater concentrations and/or point source discharge, evident at low base flow, and/or (b) increased N loading by shallow flow. The latter could result from increased atmospheric N deposition, greater annual precipitation, and/or higher surface N

concentrations imported by shallow flow. We examine these hypotheses in turn.

In 2002, the [DIN] at low base flow, which reflects that associated with deeper groundwater and point source discharge, was ~50 µM (Fulweiler and Nixon 2005) – thus lower than the value of ~65 µM observed by us – suggesting lower DIN inputs from deeper groundwater and/or point sources in 2002. This inference is supported by monitoring data of the Wood-Pawcatuck Watershed Association

from 1989 to 2017, which documented a discernible increase in riverine [$NO_3^-$] of approximately 1 µM per year at Bradford (Station 8) in the month of October, and a slighter rate of increase 0.3 µM per year in May (Figure S7a). These trends are not explained by a secular change in monthly precipitation (Figure S7b). Given that mean river discharge is generally higher in May than in October, the greater rate of increase in October suggests an increase in [DIN] of deeper groundwater entering

the river – and/or an increase in point source discharge up-river. Assuming a 15 µM difference in [DIN] during low base flow at the Stillman Bridge in 2018 compared to 2002, and a year-round discharge of deeper groundwater of $0.25 \times 10^6$ m$^3$ d$^{-1}$ (based on the mean low base flow discharge), $1.4 \times 10^6$ moles of additional DIN were potentially delivered in 2018 from the increased groundwater or point source [DIN]. This greater DIN input from deeper groundwater and/or point sources,

however, only explains a small fraction of the additional loading of $12.9 \times 10^6$ moles DIN estimated for 2018 compared to 2002.

Regional atmospheric deposition of DIN has decreased ~67 % since 2000, from ~95 µM to 36 µM DIN in 2018 (NOAA National Atmospheric Deposition Program, 2019), which should have resulted in



a lower riverine N flux given similar annual precipitation. However, 2002 was a drought year, whereas

2018 was the third wettest year on record in Washington County, RI, with total precipitation at 152

cm compared to an 80-year mean of 114 cm (NOAA National Centers for Environmental information,

2019). River discharge was thus a substantially lower in 2002, at 303 x $10^6$ $m^3$ $yr^{-1}$, compared to 702 x

$10^6$ $m^3$ $yr^{-1}$ in 2018 (Table 2). The larger riverine N loading in 2018 is thus explained by greater

precipitation and consequent discharge above low base flow, importing additional DIN (and DON)

into the river via shallow flow. Assuming a comparable [DIN] delivered by shallow flow between then

and now (26 ± 3 µmol $L^{-1}$), the greater discharge in 2018 entails an additional DIN influx of 10.1 ± 1.2

(x $10^6$) moles $yr^{-1}$, accounting for most of the estimated difference of 12.9 x $10^6$ moles DIN $yr^{-1}$

between 2018 and 2002. The greater discharge in 2018 ostensibly increased the DON influx into the

river concurrently, although the variability of our DON measurements precludes a robust estimate of

this additional flux.

In all, the greater DIN loading in 2018 compared to 2002 is explained in small part by an apparent

increase of [DIN] at low base flow – deriving from a parallel increase in groundwater [DIN] or a

potential increase in point source discharge by Kenyon Industries – and in greater part by a substantial

difference in annual river discharge and associated import of DIN (and DON).

Using our mixing curve algorithm (Equation 4), we derive an estimate of the DIN loading by the

Pawcatuck River in 2017 – a year posting a median river discharge – of 15.2 x $10^6$ moles N $yr^{-1}$ (Table

2). This estimate is 25% lower than that for 2018, yet more than double estimate for 2002,

highlighting the impact of inter-annual variability in river discharge on the annual N load.

Extrapolating DIN discharge for other years with various river flows allows for a comparison to

independent estimates of nitrogen loads from the watershed. Vaudrey et al. (2017) utilized a land-

use model to estimate the TN load from the Pawcatuck River watershed at 16.8 x $10^6$ moles TN $yr^{-1}$.

This estimate was based on 2011 land cover data, 2010 census data, precipitation from 2013 to 2015,

and included WWTF-reported data from 2011 to 2014. The land-use model thus relies on "average"

flow states and does not capture the variability in TN load associated with inter-annual variability in

river discharge. Removing the WWTF inputs occurring downstream of the Stillman Bridge allows for

a comparison to this study, resulting in an estimate of TN loading of 14.6 x $10^6$ moles TN $yr^{-1}$. Using

the mixing curve algorithm (Equation 4) and river flow during the 2013-2015 period, and further

assuming that DIN accounted for roughly half of TN as in our measurements, DIN loading is otherwise

estimated as 13.3 x $10^6$ moles DIN $yr^{-1}$ and TN loading at 26.7 x $10^6$ moles TN $yr^{-1}$ for this period. This





TN loading estimate is thus 12.1 x 10⁶ moles TN yr$^{-1}$ higher than the land-use model estimate, leaving

45 % of TN apparently unaccounted for. However, our TN measurements include both labile and non-

labile N, while the land-use model represents reactive TN and does not account for non-labile species.

On the basis that refractory humified allochthonous organic material dominates the DON pool in the

Pawcatuck River, a value of 45 % of TN being non-labile could be consistent with this system, if only

~10 % of riverine DON were labile on pertinent time scales. Seitzinger et al. (1997) otherwise

estimated that 40 – 70 % of DON from the Delaware River was reactive on pertinent time scales. The

data at hand do not permit us to resolve this quandary, although characterizing the reactivity of DON

from the Pawcatuck River is evidently crucial to mitigating eutrophication in the bay.

    Rhode Island met an ambitious goal of a 50 % reduction in N loading to Narragansett Bay in 2012

relative to 1995-1996 loads, but the Pawcatuck River was not included in these reduction priorities.

This oversight is evident in the loads we currently see to the Pawcatuck River relative to loads in rivers

draining to Narragansett Bay, located just east of the Pawcatuck River watershed. In the early 1980s

through the early 2000s, the TN load normalized to watershed area for the Woonasquatucket,

Moshassuck, Blackstone, Taunton, and Pawtuxet rivers ranged from 9.3 to 14.9 kg ha$^{-1}$ yr$^{-1}$, with an

average of 11.8 kg ha$^{-1}$ y$^{-1}$ (*as reviewed in* Narragansett Bay Estuary Program, 2017; Nixon et al., 1995;

Nixon et al., 2008). Compared to this time period, the Pawcatuck River's current load of 7.4 kg N ha$^{-1}$

yr$^{-1}$ is relatively low. However, by the late 2000s, these riverine loads were substantially reduced to

an average of 6.6 kg ha$^{-1}$ yr$^{-1}$, and have continued to decline, achieving an average of 4.8 kg ha$^{-1}$ yr$^{-1}$

in recent N budgets developed for the 2013-2015 time period (*as reviewed in* Narragansett Bay

Estuary Program, 2017; Krumholz, 2012). One exception to the success achieved in riverine loads to

Narragansett Bay is the Ten Mile River, which drains an urbanized watershed in East Providence, RI,

with a 2013-2015 load estimate of 10.2 kg ha$^{-1}$ yr$^{-1}$. However, the Ten Mile River watershed is

relatively small (~20 % of the Pawcatuck River watershed), accounting for only 7% of the riverine load

to Narragansett Bay (Narragansett Bay Estuary Program, 2017). Export from pristine temperate zones

prior to human disturbances is estimated to have been on the order 1.3 kg N ha$^{-1}$ yr$^{-1}$ (Howarth et al.

1995; 1996). Most Narragansett Bay rivers are moving toward this pristine condition, whereas the

Pawcatuck River has shown an increase in N load over time.





### 4.3.2 *Point source loading from the WWTFs and Kenyon Industries*

The Westerly and Pawcatuck WWTFs downstream of the Stillman and Westerly bridges accounted for a relatively modest fraction of the total annual nitrogen loading into the estuary, approximately 7 % (Table 2). This estimate does not consider loading from the catchment downstream of the Stillman and Westerly bridges, which would modestly lower the relative contributions of the WWTFs. Fulweiler and Nixon (2005) otherwise estimated that the WWTFs

accounted for 18 % of annual N loading into the estuary, albeit, relying on a WWTF loading estimate of $6 \times 10^6$ moles TN yr$^{-1}$, a flux notably higher than that of $1.45 \times 10^6$ moles TN yr$^{-1}$ reported by Rhode Island Department of Environmental Management and the Connecticut Department of Energy and Environmental Protection for 2002 (Vaudrey et al., 2017). Replacing the higher WWTF load in the Fulweiler and Nixon's (2005) estimate with the lower load reported by the States yields a match to

the current study, indicating that the WWTFs accounted for about 5 % of the total annual load. Independent estimates by Vaudrey et al. (2017) derived from a land-use model suggest that WWTF effluents contribute ~13 % of the riverine-plus-WWTFs TN discharged to the estuary on an annual basis, but this load included only reactive nitrogen and did not estimate the non-labile fraction measured in this and Fulweiler and Nixon's (2005) studies; including an estimate of the non-labile

fraction brings the annual contribution from WWTFs down to 8 % of the TN.

    The annual N loading into the Pawcatuck River from Kenyon Industries, as monitored by the Rhode Island Department of Environmental Monitoring from 2011-2013, was $2.7 \times 10^6$ moles TN per year, thus accounting for 6 % of the annual riverine-plus-WWTFs loading to the estuary, an input comparable to that of the WWTFs. Loading by Kenyon Industries is notable in that it is approximately

equivalent to the amount of fertilizer applied to agricultural, hay, and pasture lands throughout the whole watershed (Vaudrey et al., 2017).

    The overgrowth of nuisance macroalgae in Little Narragansett Bay is presumably fueled predominantly by nutrients delivered during warmer months, at which time riverine N loading is at a relative minimum (Table 2). While the fraction of TN loading to the estuary by the WWTFs was

negligible during colder months (< 5 %), this proportion increased to 21 % during the warmer months in 2018, from May 1st to into October 31$^{st}$. The estimated contribution from Kenyon Industries similarly increased to 16 % of total N loading during the warmer months. The influence of these point sources on algal growth during the warm season is likely to be even greater, considering that an important fraction of the total N flux from the Pawcatuck River derived from DON (38 % from May to





November), of which only a fraction may be bioavailable on pertinent time scales. Assuming a median
reactivity of riverine DON of 50 % (Seitzinger et al., 1997), the WWTFs and Kenyon industries could
account for as much as 25 and 19 % of labile N loading to the estuary during the warm season,
respectively, given a riverine DIN loading of $2.6 \times 10^6$ mol $N_{DIN}$ from May through October. Thus, we
estimate that the WWTFs contributed between 21 - 25 % of N loading to the estuary during the warm

season, and Kenyon Industries contributed 16 - 19 %.

4.4 *implications for the mitigation of eutrophication*

Our analysis suggests that the Pawcatuck River is strongly impacted by anthropogenic N input.
Compounding the problem, the drainage basin of the river is large relative to the receiving estuary,
explaining the severe eutrophication therein. The DIN concentrations and $NO_3^-$ isotope ratios indicate

substantive inputs of reactive N to the river from agricultural and/or point sources along the upper
river catchment, and from urbanized sources along the lower reach of the river. The reactive N loaded
annually into Little Narragansett Bay from the Pawcatuck River is highly influenced by the amplitude
of river discharge, increasing with discharge due to the additional import of reactive N by shallow
flow. Loading during the warmer months in 2018 was thus substantially lower than in colder months

due to lower summertime precipitation, rendering point source discharges from Kenyon Industries
and WWTFs more important to the total N loading to the estuary during the major growing season.

Reductions in summertime discharge by Kenyon Industries and the Westerly WWTF offer the
most expeditious targets to decrease N loading into the estuary, albeit, at considerable cost. The
disproportionate loading from the catchment of the upper river also begs more tempered

applications of agricultural fertilizers at adjacent turf farms and expansion of riparian buffers, in order
to effect reductions in shallow and deeper groundwater N concentrations. In the more populated
portion of the watershed, N reductions could be achieved by augmenting linkage of households to
the sewer line, transitioning traditional septic systems to advanced, N-removing septic systems, and
encouraging the dismantling of outdated, legacy cesspools (Amador et al. 2017; Narragansett Bay

Estuary Program, 2017). Within the watershed draining directly to the estuarine portion of the
Pawcatuck River south of the Westerly Bridge, 90 % of the households are connected to sewer
(Vaudrey et al., 2017). In the remainder of the watershed, where groundwater drains to a freshwater
body (wetland, pond, river) prior to entering the estuary, only 21 % of people are on sewer. This
distribution reflects the urban nature of the watershed near the coast and the more rural character





of the watershed further inland. Finally, restricting the use of lawn fertilizers and lessening the extent of impervious surfaces in and around Westerly would further aid in reducing loading from storm water.

While reductions in N loading are necessary to mitigate eutrophication in Little Narragansett Bay, target N loads have yet to be adopted by Rhode Island or Connecticut. A TN load of 50 kg ha$^{-1}_{estuary}$ yr$^{-1}$

(3.6 x 10$^3$ mol ha$^{-1}_{estuary}$ yr$^{-1}$), which is generally supportive of eelgrass, has been proposed by the scientific community (Hauxwell et al., 2003; Latimer and Rego, 2010). In 2018, the DIN and TN loads to Little Narragansett Bay were 37 x 10$^3$ and 74 x 10$^3$ moles N ha$^{-1}$ yr$^{-1}$, respectively (given a 583 ha area of estuary downstream of the Westerly Bridge), suggesting that an astounding 10 to 20-fold reduction in N loading may be required to recover eelgrass beds. We consider that a fraction of this

N load may escape the estuary directly and not be retained therein, reducing the effective annual estuarine N load. Moreover, seasonality of nitrogen delivery coupled with the warm summer growing season may point the way towards targeted summer reductions that could have greater impact on the eutrophic status of the system. Regardless, immediate mitigation efforts are necessary at this junction, not purely to realize reductions in N loading, but, more soberly, to prevent further increases

in N loading to the Pawcatuck River and continuing degradation of the river and estuary.

## 5. Conclusions

Our findings illustrate the utility NO$_3^-$ isotopologue ratios in differentiating among N sources, with implications for the management of N loading from of the watershed. In particular, the seasonal and flow-dependent nature of N loading and cycling uncovered herein presents

important considerations for mitigation efforts.

Our interpretations of NO$_3^-$ isotopologues dynamics also move beyond the traditional source attribution framework in an effort to reconcile with current theory of riverine N biogeochemistry. Nutrient spiraling theory offers a powerful conceptual basis to differentiate the influences of N sources vs. cycling on NO$_3^-$ isotopologue distributions. Continued inquiry in the context of this

framework is bound to yield novel and unexpected insights on N isotopologue cycling and, more fundamentally, on river biogeochemistry.





Table 2. Estimates of annual N loading into Little Narragansett Bay from the Pawcatuck River at Stillman Bridge

| | River Discharge $10^6$ m$^3$ yr$^{-1}$ | DIN Flux $10^6$ mol N yr$^{-1}$ | TON Flux $10^6$ mol N yr$^{-1}$ | TN Flux $10^6$ mol N yr$^{-1}$ | TN Flux $10^6$ mol N yr$^{-1}$ (May-Nov) | TN loading kg N ha$^{-1}$ yr$^{-1}$ | % of TN flux annual | % of TN flux (May-Nov) |
|---|---|---|---|---|---|---|---|---|
| **Stillman bridge (2018)** | 702 | 20.1 | 20.2 | 40.3 | 4.8 | 7.4[§] | – | – |
| **Kenyon Industries** | – | – | – | 2.7[*] | 1.0[€] | – | 6 | 16 |
| **Westerly WWTF[¶]** | – | 1.4 | 1.1 | 2.5 | 1.2 | – | 6 | 20 |
| **Pawcatuck WWTF[†]** | – | – | – | 0.3 | <0.1 | – | <1 | 2 |
| **Stillman bridge (2002)[‡]** | 303 | 7.2 | 6.2[^] | 16.0 | – | 3.1[§] | – | – |
| **Land use model[c]** | – | – | – | 17.8 | – | 3.6[¥] | – | – |
| **Mixing model (2017)[£]** | 516 | 15.2 | – | – | – | – | – | – |

(inclusive of Kenyon Industries) and from the Westerly and Pawcatuck Waste Water Treatment Facilities
downstream.

[§]Based on a watershed area of 760 km$^2$; [*]DEM-monitored loading in 2012. [€]Permitted seasonal loading;
[¶]Measured; [†]Reported; [‡]Fulweiler & Nixon (2005);[^]as DON [c]Vaudrey (2016); [£]Eq. 4; [¥]Based on a watershed area of
660 km$^2$ established from ArcHydro.



*Data availability.* All data were submitted to Rhode Island Department of Environmental
Management and the Connecticut Department of Energy and Environmental Protection, and
will also be archived online in the PANGAEA data repository.

*Author contributions.* VRR and JG conceive the research question, designed the study approach,
led the field survey, ensured data curation and conducted formal analysis. SC, MLB, CPK, LAT, and
HCW assisted with data collection and analysis. CMM assisted with statistical analyses. CRT and
HH provided use of specialized facilities. JG and JMPV secured funding for the investigation. VRR
and JG wrote the first draft of the paper, and all co-authors contributed to writing review and
editing.

*Competing interests.* The authors declare that they have no conflicts of interest.

*Acknowledgements.* We thank Clare Schlink, Reide Jacksin, Lindsey Potts, Danielle Boshers-Snow,
Anna Alvarado, Peter Ruffino and Matt Lacerra for assistance with fieldwork and/or laboratory
analyses. We are also grateful to Nicholas De Gemmis and the Jacobs Group at Westerly Waste
Water Treatment facility for providing us with weekly samples. Water quality monitoring by the
Wood-Pawcatuck Watershed Association provided important historical data that facilitated our
interpretations. Monitoring of water quality in Little Narragansett Bay by Clean Up Sound and
Harbors (CUSH) inspired our effort to determine sources of nutrients to the estuary.

*Financial support.* This work was funded by a Connecticut Sea Grant award to JG and JMPV under
NOAA award NA18OAR4170081, project R/ER-30; and an NSF CAREER award to JG (OCE-
1554474).

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
