# Peer review of "Seasonality of nitrogen sources, cycling and loading in a New England river discerned from nitrate isotope ratios"

_Biogeosciences, 2020_

## Referee Comment (RC1) · Anonymous Referee #1 · 13 Nov 2020

Review of " Sources and Cycling of nitrogen in a New England river discerned from nitrate isotope ratios" by Rollinson et al.

Summary

Rollinson and colleagues present a comprehensive examination of nitrogen (N) loading dynamics in a New England watershed. The analysis includes measurements of DIN including nitrate (NO3-), nitrite (NO2-) and ammonium (NH4+) as we all as dissolved organic (DON) and particulate (PN) forms. In contrast to previous studies of N loading in this watershed, the authors also leverage the use of nitrate N and O isotopes for constraining confounding influences of source mixing and cycling mechanisms. Together

with various temporal and spatial perspectives (seasonal transects of the whole river, weekly site sampling, river discharge measures and point source characterization), they stitch together a comprehensive picture of N sources and controls on loading from the watershed into the estuary. The study indicates very little uncycled atmospheric N loading and identifies underlying drivers of N loading that stem from differing hydrologic regimes (e.g., base flow conditions vs. shallow flow influences). They authors also outline how nitrate N and O isotopes are expected to behave in the framework of 'nutrient spiraling' – and use the differential behaviors of N and O isotopes to constrain cycling and source partitioning.

Major comments

Overall, the manuscript is very well-written and covers a lot of ground. The data are of high quality and the analysis and interpretation of the data is sound.

One criticism I have is that the manuscript is probably overly long-winded in some aspects and might benefit from some trimming and tightening to make it more approachable to a broader audience. I did appreciate the application of the data to the broader understanding of sources and cycling phenomena in the watershed and the thoroughness of this discussion (sections 4.2), but thought that the discussion of loading (4.3), for example, could be condensed.

My only other critique is that there were times when I was left wondering about the error on some of the endmember estimates and flux terms. My guess is that the small distinctions in average endmember isotopic compositions might be overwhelmed by natural variability in sample population (and/or in the intercept on the modified Keeling plot)?

Also, for example, it is not clear how 'close' the flux comparison between the 2018 data in this study may compare the historical 2002 data from Fulweiler and Nixon (Lines 680 to 689). While it seems clear that the drastically disparate hydrologic regimes of the two years underlie the major changes in N fluxes, having a better understanding

of the magnitude of the error associated with these watershed scale flux estimates would help readers put the assessment in to a clearer context. Similarly, the estimates of endmember concentrations (L692) should have some indication of confidence. By some measures, 50 and 65 are not all that different, for example. Finally, the same can be said for the estimates of endmember $\delta15N$ and $\delta18O$ values (for example, L457; L490 to L500).

To be clear I am not trying to insinuate that the data and findings are suspect in any way – just that more attention could be given to presenting the error intrinsic to such endmember estimation.

Minor comments L99: Sigman et al 2019 reference – missing?

L101: what is meant by 'inherent' cycling?

L265: 'barring a single outlying value...'

L303: were highly anti-correlated (?)

It might be useful to understand whether the WWTFs are of the combined-sewer overflow type or not, which plays into the residence time of waste in the facilities – and hence N speciation. There is a clear seasonality in the WWTF speciation data that is not really highlighted or addressed. So, while the DIN and TON fluxes from the WWTF are remarkably constant, the NH4+ and NO3- fluxes would not necessarily be constant. It isn't clear whether this really plays into any of the overall findings.

It appears that the WWTF samples were not analyzed for nitrate isotopes? This would be a unique dataset and could offer some interesting insights. Given the large shifts in total N speciation in the WWTF – I suspect there would be some substantial variation in the N and O isotopes of WW effluent as well. While these data are not necessarily paramount to the conclusions presented in the paper, knowing more about isotopic variability associated with annual WWTF operations and effluent would be valuable to the riverine N biogeochemistry community in general. Side question – was nitrite

measured or reported from WWTF samples?

L377:. . . would admittedly arise from loading by point sources to the degree that a point source has elevated conductivity.

L456: Refer to Figure 8.

L498: ". . . values of NO3- in rainwater."

L500: . . . with higher discharge can thus be partially explained. . .

L507: Kendall is misspelled.

L516 to L522. Consider splitting this sentence into two sentences.

L529-531: Please include reference here.

L542: which limited light penetration.

L620: to primarily reflect

L640: there was little to no accumulated snow in March 2019

L645: I think it would be good to state that no samples were taken from Kenyon Industries much earlier – when it is introduced as a potential point source.

---

## Referee Comment (RC2) · Anonymous Referee #2 · 23 Dec 2020

The authors present a comprehensive study about the nitrogen sources and cycling in a well-studied river and estuary system in New England, USA. They used a complex one-year data set, including all nitrogen components such as DIN (ammonium, nitrite and nitrate), DON and PN, additional the stable isotopes (15N and 18O) of nitrate including 17N for deposition analysis. The data set contains weekly data from two station, seasonal transects in the river at 15 station, one high resolution short transect, and data from two WWTP near the mouth of the river. Additional, ammonium and nitrate concentrations and stable isotope of nitrate in atmospheric deposition were provided. The main findings are that nitrate sources mainly stemming from the groundwater and from shallower groundwater and surface flow during higher river discharge

during the cold month. WWTP and industrial zone runoffs had a portion of approx. 20 %. The stable isotope analysis suggest the river-in nitrogen cycling is not that important in relation to the sources. One suggestion is that by nutrient spiraling occurring were reverse processes are not visible in concentrations or isotope signals. In comparisons to former studies in the river estuary system, there is an increase of nitrogen loads to the adjacent Little Narragansett Bay, which causes eutrophication.

The manuscript is well written and presented a robust data set and the interpretation and discussion based on that and is not excessive, although, the text is somehow to long and can be shorted, especially in the discussion. The study present a good contribution to the discussion about the role of rivers and estuaries nutrients transport from the land to the ocean.

Nevertheless, I have some comments and questions on some issues in the manuscript.

L 11ff: The abstract is a bit too long and should focus more on the main finding. It seems like a list of what were done and what were discussed. Be more specific.

L 84: The 18O values of nitrate produced by nitrification ($\leq 1\ldots$) is misleading, because later on you discuss it is bit different way, that is depending on the 18value of the water and so 1‰ higher. (see L 595 ff)

L 133: Maybe, just because I'm not a native speaker and not from USA. What exactly are turf farms? Do they produce grass, which can put later in the garden or it is something to produce peat. This could explain the high concentration of tannin in the water.

Figure 1, L144: I needed a bit to understand the description and the map, please present the sampling sites and map in a clearly arranged way.

L 150 Explain shorty why you are not measure in the same period and be aware that deposition data are just represent a period of higher precipitation

L 180. Do you also measure 15N NH4 isotopes? Could be interesting to see what

happened to the deposited nitrogen in whole...

Figure 2: the figure is relatively small. I'm surprised that the summer nitrate concentration were higher in summer than in winter.

Figure 4: In "a" and "b" your present the fluxes from the WWTP. The discharge from the WWTP was much more smaller that the discharge of river itself, so that the presentation is a bit misleading, especially because later on the discussion of the source count the WWTP later on (L770 ff)

Figure 6: Explain why the station 6 is separated (Tributary)

Before L366. The depositions results are not presented in the results section, but later in the discussion.

L400: Reference for the tannins are missing. How high is the refractory and labile part of the DON

L403: The unexpected nitrate concentration in summer should be compared to other rivers like you already done with the results from Fulweiler&Nixon

L 526: What happened with the NH4 in the atmospheric deposition?

L661: Use these turf farms a high amount of fertilizers?

L 685: Who or what is responsible. Agriculture? Too much fertilizers? Turf farms?

L 703: What is the main souce for the increase of nitrate in the groundwater? I would expect higher use of fertilizer?

—————————————————

---

## Author Comment (AC1) · 22 Jan 2021

Rollinson et al. Anonymous Referee #1

 Review of "Sources and Cycling of nitrogen in a New England river discerned from nitrate isotope ratios" by Rollinson et al.

Summary Rollinson and colleagues present a comprehensive examination of nitrogen

(N) loading dynamics in a New England watershed. The analysis includes measurements of DIN including nitrate (NO3-), nitrite (NO2-) and ammonium (NH4+) as we all as dissolved organic (DON) and particulate (PN) forms. In contrast to previous studies of N loading in this watershed, the authors also leverage the use of nitrate N and O isotopes for constraining confounding influences of source mixing and cycling mechanisms. Together with various temporal and spatial perspectives (seasonal transects of the whole river, weekly site sampling, river discharge measures and point source characterization), they stitch together a comprehensive picture of N sources and controls on loading from the watershed into the estuary. The study indicates very little uncycled atmospheric N loading and identifies underlying drivers of N loading that stem from differing hydrologic regimes (e.g., base flow conditions vs. shallow flow influences). They authors also outline how nitrate N and O isotopes are expected to behave in the framework of 'nutrient spiraling' – and use the differential behaviors of N and O isotopes to constrain cycling and source partitioning.

Major comments Overall, the manuscript is very well-written and covers a lot of ground. The data are of high quality and the analysis and interpretation of the data is sound.

One criticism I have is that the manuscript is probably overly long-winded in some aspects and might benefit from some trimming and tightening to make it more approachable to a broader audience. I did appreciate the application of the data to the broader understanding of sources and cycling phenomena in the watershed and the thoroughness of this discussion (sections 4.2), but thought that the discussion of loading (4.3), for example, could be condensed. We concur with the Reviewer's assessment. We have condensed section 4.3.

My only other critique is that there were times when I was left wondering about the error on some of the endmember estimates and flux terms. My guess is that the small distinctions in average endmember isotopic compositions might be overwhelmed by natural variability in sample population (and/or in the intercept on the modified Keeling plot)? Also, for example, it is not clear how 'close' the flux comparison between
the 2018 data in this study may compare the historical 2002 data from Fulweiler and Nixon (Lines 680 to 689). While it seems clear that the drastically disparate hydrologic regimes of the two years underlie the major changes in N fluxes, having a better understanding of the magnitude of the error associated with these watershed scale flux estimates would help readers put the assessment in to a clearer context. We consider that our N flux terms are well constrained, as these derive from highly resolved discharge measurements (from the river flux gauges), and temporally well-resolved concentration estimates (weekly). We imposed some statistical rigor to the flux estimates by using Beale's ratio estimator, which is cited to provide high estimation accuracy and relatively low bias of total nitrogen and nitrate (e.g., Lee et al. 2016; J. Hydrol). Indeed, the bias correction factor was 1 for all computations (i.e., the term in parentheses in Eq. 3a). Nevertheless, to approximate the potential uncertainty in the flux estimates, we conducted an additional bootstrap of the N flux estimates, and posted the resulting standard deviation of respective terms in Table 2. We added text to the methods to describe the bootstrapping exercise. The resulting uncertainty of the flux estimates is relatively modest, particularly for DIN (<10%), less so for DON and corresponding TN ($\leq$17%). In light of this uncertainty, flux estimates remain distinct between 2018 and 2001.

Similarly, the estimates of endmember concentrations (L692) should have some indication of confidence. By some measures, 50 and 65 are not all that different, for example. Finally, the same can be said for the estimates of endmember _15N and _18O values (for example, L457; L490 to L500).

The reviewer raises a valid point. The low-flow [NO3-] end-member was "guesstimated" and devoid of confidence estimates. To remedy this, we derived the low-flow end-member [NO3-] from the best fit of Equation 4 to the observations, stipulating a low base flow asymptote of $2.2 \times 10^8$ L d-1 (the lowest observed discharge in 2018) and a concentration increase of 26 $\mu$moles per L, yielding an end-member value of 64 $\pm$ 9 $\mu$M.

While we did not obtain the data from the Fulweiler & Nixon study for direct comparison; nevertheless, the low flow [NO3-] reported therein is undeniably lower in 2002 than in 2018.

With regard to the isotopic end-members, we now report the standard error for the intercepts of the modified Keeling Plots, which are relatively small. We considered performing an additional jackknife of the regression analyses, but concluded that the interpretations of the study are not contingent on the absolute values of the intercept, only whether these were higher or lower than observed at low base flow.

To be clear I am not trying to insinuate that the data and findings are suspect in any way – just that more attention could be given to presenting the error intrinsic to such endmember estimation.

Minor comments L99: Sigman et al 2019 reference – missing? Corrected to Sigman and Fripiat, 2019. L101: what is meant by 'inherent' cycling? The word was unnecessary. We removed it for simplification.

L265: 'barring a single outlying value. . .' We changed to "notwithstanding a single outlying value."

L303: were highly anti-correlated (?) We changed to ". . .were similar in grab vs. composite samples." It might be useful to understand whether the WWTFs are of the combined-sewer overflow type or not, which plays into the residence time of waste in the facilities – and hence N speciation. There is a clear seasonality in the WWTF speciation data that is not really highlighted or addressed. So, while the DIN and TON fluxes from the WWTF are remarkably constant, the NH4+ and NO3- fluxes would not necessarily be constant. It isn't clear whether this really plays into any of the overall findings. It appears that the WWTF samples were not analyzed for nitrate isotopes? This would be a unique dataset and could offer some interesting insights. Given the large shifts in total N speciation in the WWTF – I suspect there would be some substantial variation in the N and O isotopes of WW effluent as well. While these data are

not necessarily paramount to the conclusions presented in the paper, knowing more about isotopic variability associated with annual WWTF operations and effluent would be valuable to the riverine N biogeochemistry community in general. Side question – was nitrite measured or reported from WWTF samples? The Westerly-WWTF does not have combined sewer overflow. As pointed out by the reviewer, we neglected to discuss the N speciation and associated dynamics of the W-WWTF effluent, out of concern for the length of the manuscript. And although we originally intended to, we ultimately did not conduct isotope ratio analyses of nitrate in the effluent, deciding these were not central to our study. That said, we intend to further exploit the WWTF data and measure the isotope ratios in our collected effluent samples for an upcoming study of N cycling within the estuary proper.

L377:. . . would admittedly arise from loading by point sources to the degree that a point source has elevated conductivity. Not necessarily. The conductivity of deeper groundwater is generally higher than that of shallow groundwater, because it is in contact with the substrate and bedrock for longer. One could envision groundwater that has a lower nutrient concentration than shallower groundwater, such that nutrients from a point source are less diluted at low base flow. That said, we do not think this is the case, but cannot rule out this possibility entirely.

L456: Refer to Figure 8. The line refers to Figure 2, which was described in the Results, such that it need not be re-iterated in this part of the discussion, where we summarize salient results.

L498: ". . . values of NO3- in rainwater." We added this qualifier to the text.

L500: . . . with higher discharge can thus be partially explained. . . We modified the clause to ". . . The increase in $\delta 18ONO3$ with increasing discharge. . ."

L507: Kendall is misspelled. Fixed.

L516 to L522. Consider splitting this sentence into two sentences. We split it into two

sentences.

L529-531: Please include reference here. We added a reference.

L542: which limited light penetration. Fixed.

L620: to primarily reflect Added "primarily."

L640: there was little to no accumulated snow in March 2019 Fixed.

L645: I think it would be good to state that no samples were taken from Kenyon Industries much earlier – when it is introduced as a potential point source. We added this information at line 161 in the Methods.

Please also note the supplement to this comment:
https://bg.copernicus.org/preprints/bg-2020-390/bg-2020-390-AC1-supplement.pdf

---

## Author Comment (AC2) · 22 Jan 2021

data from two WWTP near the mouth of the river. Additional, ammonium and nitrate concentrations and stable isotope of nitrate in atmospheric deposition were provided. The main findings are that nitrate sources mainly stemming from the groundwater and from shallower groundwater and surface flow during higher river discharge C1 during the cold month. WWTP and industrial zone runoffs had a portion of approx. 20 %. The stable isotope analysis suggest the river-in nitrogen cycling is not that important in relation to the sources. One suggestion is that by nutrient spiraling occurring were reverse processes are not visible in concentrations or isotope signals. In comparisons to former studies in the river estuary system, there is an increase of nitrogen loads to the adjacent Little Narragansett Bay, which causes eutrophication. The manuscript is well written and presented a robust data set and the interpretation and discussion based on that and is not excessive, although, the text is somehow to long and can be shorted, especially in the discussion. The study present a good contribution to the discussion about the role of rivers and estuaries nutrients transport from the land to the ocean.

Nevertheless, I have some comments and questions on some issues in the manuscript.

L 11ff: The abstract is a bit too long and should focus more on the main finding. It seems like a list of what were done and what were discussed. Be more specific. We agree with the reviewer's assessment. We modified the title of our study to make it more descriptive and pointed, and we trimmed down and re-arranged and modified the abstract to emphasize the more salient points of the study.

L 84: The 18O values of nitrate produced by nitrification (_1. . .) is misleading, because later on you discuss it is bit different way, that is depending on the 18value of the water and so 1‰ higher. (see L 595 ff) We modified this text to specify that the ïĄd'18O of NO3- produced by nitrification aligns closely with that of ambient water.

L 133: Maybe, just because I0m not a native speaker and not from USA. What exactly are turf farms? Do they produce grass, which can put later in the garden or it is something to produce peat. This could explain the high concentration of tannin in the

water. Turf farms are a characteristically North American enterprise, producing mats of mono-specific grass for suburban homes, which then require an obscene amount of herbicides, pesticides, water and fertilizers to maintain. They are not used for peat production. The elevated tannins in the river presumably derive from the forest soils and litter.

Figure 1, L144: I needed a bit to understand the description and the map, please present the sampling sites and map in a clearly arranged way. We extended the figure caption to be more explanatory.

L 150 Explain shorty why you are not measure in the same period and be aware that deposition data are just represent a period of higher precipitation We are aware that our precipitation samples only cover the fall season. The samples were opportunistic, given to us by an undergraduate student studying mercury deposition. Hence, the data do not span the whole of our study. While the attribution of the % atmospheric nitrate may have been more nuanced seasonally given a broader span of measurements, differences would likely be within the error of attribution ($\pm$ 1‰ and, more importantly, would in no way change the inferences drawn from our data – that the fraction of uncycled atmospheric nitrate in the river was negligible.

L 180. Do you also measure 15N NH4 isotopes? Could be interesting to see what happened to the deposited nitrogen in whole. . . We did not measure the ammonium N isotope ratios in rainwater, as this was outside the scope of our study, and would have required an altogether different (and very exacting) analytical technique (see Zhang et al. 2007).

Figure 2: the figure is relatively small. I0m surprised that the summer nitrate concentration were higher in summer than in winter. The figure is not at its final size (we think). The dynamic we observed here has been observed in other mid-latitude streams and rivers (e.g., Mulholland et al. 1997).

Figure 4: In "a" and "b" your present the fluxes from the WWTP. The discharge from the

WWTP was much more smaller that the discharge of river itself, so that the presentation is a bit misleading, especially because later on the discussion of the source count the WWTP later on (L770 ff) We disagree with the reviewer's assessment. Figure 4, upon close inspection, reveals that the N flux from the WWTF is negligible during high discharge, but relatively important in summer during low river discharge. What was remarkable to us is that the facility maintains a constant N flux throughout the year, in spite of seasonal differences in constituents (Figure 5).

Figure 6: Explain why the station 6 is separated (Tributary) We have added this explanation in text. Station 6 is in a tributary feeding the main study river rather than in the river proper. Before L366. The depositions results are not presented in the results section, but later in the discussion. The deposition results are now alluded to in the Results at Line 274.

L400: Reference for the tannins are missing. How high is the refractory and labile part of the DON? Good question. We added a reference. Tannins are considered recalcitrant, and we added a reference which validates this assertion.

L403: The unexpected nitrate concentration in summer should be compared to other rivers like you already done with the results from Fulweiler&Nixon. We feel that this would add to an already long manuscript that we have struggled to shorten. Readers interested in regional differences can delve of their own initiative from the cited reference (Narragansett Bay Estuary Program, 2017). Importantly, the more elevated concentration in summer is not an entirely novel dynamic, as it is germane to other systems (e.g., Mulholland et al. 1997).

L 526: What happened with the NH4 in the atmospheric deposition? While that is not a focus of the study, NH4+ in aerobic soils and streams can be assimilated by plants and algae or nitrified to nitrate, which can be denitrified in hypoxic and anoxic soils (e.g., hydrated soils). That there typically is little to no uncycled atmospheric nitrate in mid-latitudes rivers (e.g., Sebestyen et al. 2019; Mengis et al. 2001) suggests that

reactive N deposited at the land surface is quickly cycled.

L661: Use these turf farms a high amount of fertilizers? Typically, yes.

L 685: Who or what is responsible. Agriculture? Too much fertilizers? Turf farms? We hoped to have constraints on specific areas of N loading from the study. All that we can conclude is that a disproportionate fraction of the polluting N occurs up-river in the agricultural part of the watershed (∼30%; Line 663 of total), and down-river (anywhere from 20 to 50% is our best estimate; Line 675). Up-river, we hypothesize that turf farms contribute significantly to N loading, but there is also a point source (Kenyon Industries) that looks to be important, as detailed in the discussion.

L 703: What is the main source for the increase of nitrate in the groundwater? I would expect higher use of fertilizer? We don't know. It could be the proliferation of turf farms, increased fertilizer use by farms and households, and/or increased loading by Kenyon Industries.